# PromptWizard:
# Task-Aware Prompt Optimization Framework

## Abstract

Large language models (LLMs) have transformed AI across diverse domains, with *prompting* being central to their success in guiding model outputs. However, manual prompt engineering is both labor-intensive and domain-specific, necessitating the need for automated solutions. We introduce PromptWizard, a novel, fully automated framework for discrete prompt optimization, utilizing a self-evolving, self-adapting mechanism. Through a feedback-driven critique and synthesis process, PromptWizard achieves an effective balance between exploration and exploitation, iteratively refining both prompt instructions and in-context examples to generate human-readable, task-specific prompts. This guided approach systematically improves prompt quality, resulting in superior performance across 45 tasks. PromptWizard excels even with limited training data, smaller LLMs, and various LLM architectures. Additionally, our cost analysis reveals a substantial reduction in API calls, token usage, and overall cost, demonstrating PromptWizard's efficiency, scalability, and advantages over existing prompt optimization strategies.

## 1 Introduction

Large language models (LLMs) like GPT-4 (OpenAI et al., 2024) have achieved remarkable performance across diverse tasks (Colombo et al., 2024; Nguyen et al., 2023; Zhang et al., 2024). At the core of this success is *prompting*—the process of providing input instructions to guide models toward desired outputs. Studies have shown that prompting significantly influences LLM performance, making *prompt engineering*—the design and refinement of prompts—critical for maximizing accuracy (Wang et al., 2023c;b; Nori et al., 2023). However, crafting effective prompts remains a labor-intensive and domain-specific task, requiring human expertise and subjective judgment. As models evolve and tasks vary, the need to repeatedly design prompts raises an important question: *Can prompt engineering be automated to streamline this process and enhance scalability?*

Automatically generating optimal prompts is a key challenge in the era of LLMs (Pryzant et al., 2023; Zhou et al., 2023). Some approaches, such as gradient-based methods, have been used to optimize prompts by leveraging token probabilities and model gradients (Deng et al., 2022; Zhang et al., 2022a). However, these methods are limited to white-box (open-source) models, as they require direct access to the model's internal mechanics (Liu et al., 2023). The most powerful LLMs today, like GPT-4 and Gemini, are typically black-box (closed-source) and accessible only through APIs, making such techniques impractical and are often resource-intensive.

This necessitates gradient-free prompt optimization strategies. Recent methods have focused on enumerating diverse prompts or refining existing ones to optimize instructions for black-box LLMs (Zhou et al., 2023; Lin et al., 2024; Chen et al., 2023; Fernando et al., 2023; Guo et al., 2024). These strategies can be broadly classified into two types: *continuous* and *discrete* prompt optimization. **Continuous approaches**, like InstructZero (Chen et al., 2023) and Instinct (Lin et al., 2024), convert prompt optimization into a continuous problem by using soft prompts. These soft prompts are fed to open-source LLMs to generate instructions, which are then evaluated by the target black-box LLM. The feedback is used to train a Bayesian optimizer (BO) or neural network (NN) to predict better instructions. However, these methods require additional training of NNs and their performance often varies based on the open-source model and task complexity. For more complex tasks, learning the optimal prompt-performance mapping becomes challenging. On the other hand, **discrete methods** like PromptBreeder (Fernando et al., 2023) and EvoPrompt (Guo et al., 2024) generate multiple prompt

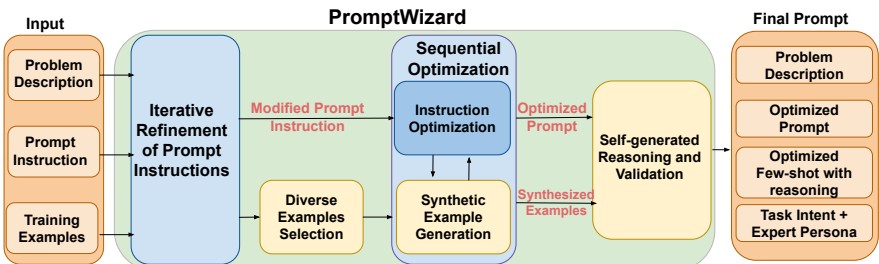

Figure 1: Overview of `PromptWizard` framework.

versions using evolutionary or self-referential strategies. While these methods expand exploration by scoring prompts, they lack feedback mechanisms, leading to inefficient and suboptimal exploration.

In this paper, we propose `PromptWizard` (PW), a discrete prompt optimization framework for black-box LLMs. `PromptWizard` employs a *self-evolving* mechanism where the LLM generates, critiques, and refines its own prompts and examples, continuously improving through iterative feedback and synthesis. This *self-adaptive* approach ensures holistic optimization by evolving both the instructions and in-context learning examples for better task performance. `PromptWizard` operates in two phases: (i) *Prompt generation (one-time)*, where it processes a high-level problem description and training samples, using LLMs to mutate, score, critique, synthesize, reason, and validate prompts and examples; (ii) *Inference (test-time)*, where the final optimized prompt and examples are applied to test samples.

PW's approach follows a structured strategy (See Figure 1): ❶ First, starting with a problem description and initial prompt instruction, PW generates variations of the instruction by prompting LLMs to mutate it. Based on performance, the best prompt is selected. Unlike uncontrolled evolutions in prior methods (Fernando et al., 2023; Guo et al., 2024), PW incorporates a critique component that provides feedback, thus guiding and refining the prompt over multiple iterations. ❷ Unlike other discrete approaches, PW also optimizes in-context examples. PW selects a diverse set of examples from the training data, identifying positive and negative examples based on their performance with the modified prompt. Negative examples help inform further prompt refinements. ❸ Examples and instructions are sequentially optimized, using the critique to generate synthetic examples that address the current prompt's weaknesses. These examples are integrated to further refine the prompt. ❹ PW generates detailed reasoning chains via Chain-of-Thought (CoT), enriching the prompt's capacity for problem-solving. ❺ PW aligns prompts with human reasoning by integrating task intent and expert personas, enhancing both model performance and interpretability.

Our work distinguishes itself from previous approaches in several key aspects: **1. Guided Exploration:** `PromptWizard` introduces a feedback-driven critique-and-synthesis mechanism, refining prompts based on performance insights. This guided *exploration* systematically improves prompt quality, overcoming the randomness and inefficiencies in methods like PromptBreeder (Fernando et al., 2023), OPRO (Yang et al., 2024), and EvoPrompt (Guo et al., 2024)(Section 3.1). **2. Sequential Optimization of Instructions and Examples:** `PromptWizard` dynamically and iteratively optimizes both prompt instructions and in-context examples in tandem, outperforming methods that optimize these components in isolation. This strategy allows deeper *exploitation* of task-specific nuances, leading to superior prompt quality (Section 3.3). **3. Efficient Example Synthesis & Error Analysis:** `PromptWizard` enhances efficiency by utilizing a compact set of diverse examples (up to 25) and leveraging error-driven self-reflection to generate synthetic examples. Combined with Chain-of-Thought reasoning, this approach offers robust and scalable prompt refinement, setting it apart from existing methods (Section 3.4).

We evaluate the effectiveness of `PromptWizard` on the widely-used Big Bench Instruction Induction (BBII), Big Bench Hard (BBH), and arithmetic reasoning datasets, covering over 45 tasks ranging from general reasoning to domain-specific challenges (Section 4). As shown in Figure 2, `PromptWizard` consistently outperforms state-of-the-art approaches, including Instinct, InstructZero, APE, PromptBreeder, and EvoPrompt on the BBII dataset.

Through extensive experimentation, we demonstrate that `PromptWizard` consistently outperforms SOTA baselines in both zero-shot and few-shot scenarios, while maintaining superior efficiency (Section 5.1). Our comprehensive cost analysis highlights the significant reduction in

API calls, token usage, and overall expenses, showcasing PW's ability to deliver high-quality prompts with minimal computational cost (Section 5.2). Furthermore, we conduct numerous experiments to showcase `PromptWizard`'s efficacy with limited training data and smaller LLMs, along with ablation studies that assess its performance across different base LLMs (Section 6).

Our main contributions are: (i) we introduce `PromptWizard`, a novel framework for automatic discrete prompt optimization using a self-evolving, self-adapting mechanism. Through feedback-driven critique and synthesis process, PW strikes an effective balance between exploration and exploitation, iteratively refining both prompt instructions and in-context examples. Thus generating human-readable, task-specific prompts, (ii) we demonstrate PW's superior performance and efficiency across 45 tasks, outperforming SOTA methods.

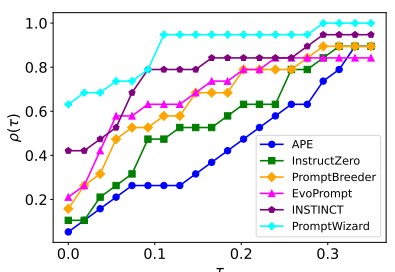

Figure 2: Performance profile curve of `PromptWizard` over other baselines (Section 5.1, Appendix 11).

## 2 RELATED WORK

Research in prompt optimization has increasingly shifted toward automating prompt creation due to the limitations of handcrafted prompts (Moradi & Samwald, 2021; Madaan & Yazdanbakhsh, 2022; Wei et al., 2022). Recent work has introduced various techniques for automating prompt generation, broadly classified into continuous and discrete (Yang et al., 2024; Guo et al., 2024). Below, we examine these methods, their limitations, and how PromptWizard (PW) advances the field.

**Continuous Prompt Optimization.** Continuous methods, such as InstructZero (Chen et al., 2023) and Instinct (Lin et al., 2024), treat prompt optimization as a continuous learning problem using soft prompts—trainable vectors that fine-tune responses from open-source LLMs. These soft prompts are used to generate responses, with feedback guiding the optimization through models like Bayesian optimizers or neural networks. While flexible, these methods face several key limitations: (i) They require additional neural network training, leading to high computational costs, (ii) Their adaptability to complex tasks that need nuanced prompts is limited, as soft prompts are not human-interpretable and struggle to capture the depth of task-specific reasoning, (iii) For more intricate tasks, such as arithmetic reasoning, mapping the relationship between prompt structure and performance becomes challenging, often leading to suboptimal or inconsistent results. Thus, while continuous methods improve prompt generation, their scalability and interpretability in complex tasks remain non-trivial.

**Discrete Prompt Optimization.** Discrete methods focus on exploration by generating multiple prompt versions and selecting the best among candidates. These methods rely on strategies like Monte Carlo searches or evolutionary processes. For example, APE (Zhou et al., 2023) iteratively proposes and selects optimal prompts through a Monte Carlo search, while PromptBreeder (Fernando et al., 2023) mutates prompts using different thinking styles, evolving prompts in a self-referential manner. Other methods, such as OPRO (Yang et al., 2024) and EvoPrompt (Guo et al., 2024), rely on prompt mutations, evolutionary algorithms and evaluations on fixed training samples. However, discrete methods have notable drawbacks: (i) They are often query-inefficient due to their reliance on local search techniques, which fail to balance exploration and exploitation effectively, (ii) These methods tend to explore the prompt space randomly or through mutations without a structured mechanism for feedback, resulting in suboptimal and unguided refinement of prompts. Recent methods optimize both instructions and examples in prompting, emphasizing the importance of example selection through random or diversity-based or adversarial techniques (Do et al., 2024; Wan et al., 2024). In contrast, PW uses a LLM to analyze and synthesize examples, dynamically enhancing prompt quality and outperforming traditional fixed-criteria strategies.

**Comparison and Motivation for PromptWizard.** `PromptWizard` (PW) advances beyond these limitations by introducing a self-evolving and self-adaptive mechanism that better balances exploration and exploitation. Unlike prior methods, PW utilizes a feedback-driven critique-and-synthesis process, which iteratively refines both prompt instructions and in-context examples. This feedback loop, guided by performance insights, leads to more systematic and efficient exploration compared to random or mutation-based strategies like those employed by PromptBreeder and EvoPrompt. Key advantages of PW include: (i) Deeper Exploitation of Task Nuances: By optimizing prompts and examples together, PW can capture the nuanced requirements of complex tasks that continuous and discrete methods often miss, (ii) Human-Interpretable and Scalable: Unlike soft prompts, PW

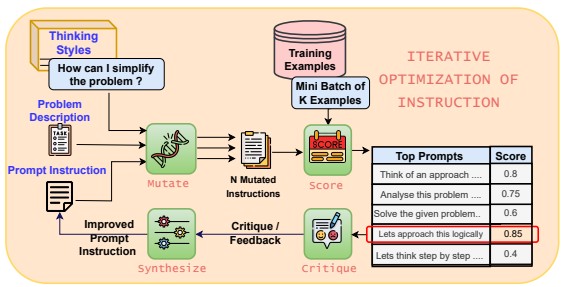
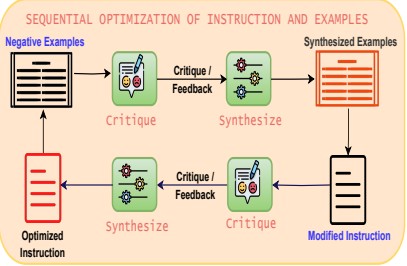

Figure 3: Iterative Optimization of Prompt Instruction.   Figure 4: Sequential Optimization.

generates human-readable prompts that align with task intent, making it more interpretable and easier to scale across diverse applications, (iii) Efficiency: PW is significantly cost-efficient, reducing the number of API calls and token usage while delivering superior performance. Evaluated across over 45 complex tasks, PW consistently outperforms state-of-the-art approaches, such as Instinct, InstructZero, APE, EvoPrompt and PromptBreeder.

In summary, PW advances prompting by addressing the exploration-exploitation trade-off more effectively than prior approaches, delivering higher-quality prompts with less computational overhead.

## 3  PROMPTWIZARD FRAMEWORK

We introduce `PromptWizard` (PW), a general-purpose framework designed to optimize prompts through a self-evolving and self-adapting mechanism (see Figure 1). PW harnesses the capabilities of LLMs to iteratively synthesize, critique, and refine both prompt instructions and in-context examples, tailoring them to specific tasks across diverse domains. The five key steps are described next.

**Problem Formulation.** In our approach, we start with an initial prompt instruction $P$ e.g., "Let's think step by step to arrive at the solution of this mathematical problem"), along with a problem description and a set of training samples represented as $(Q, A) = \{(q_i, a_i)\}_{i=1}^N$, where $q_i$ and $a_i$ are input-output pairs (questions and answers). The LLM model $L$ generates outputs with probabilities $p_l(a_i \mid q_i, P, a_f, q_f)$, where $q_f$ and $a_f$ are the few-shot examples. The goal of `PromptWizard` is to iteratively optimize both the prompt and the few-shot examples to maximize task accuracy $A$, which represents the model's performance on the target task. The refined prompt $\hat{P}$ should improve the model's ability to generate accurate outputs.

### 3.1  ITERATIVE REFINEMENT OF PROMPT INSTRUCTIONS

The first step of the `PromptWizard` framework focuses on refining prompt instructions through a systematic, feedback-driven process. This ensures the prompt evolves in a targeted way, addressing specific task needs while avoiding unnecessary changes (see Figure 3).

1. **MutateComponent:** PW starts with an initial problem description and generates prompt variations using predefined cognitive heuristics or thinking styles. These heuristics guide the LLM to create diverse perspectives on the problem, ensuring varied and rich prompt instructions. For example, the thinking styles might encourage questions like "How can I simplify the problem?" or "What alternative perspectives exist?" This targeted generation of mutations improves the diversity of prompt instructions compared to random approaches. By using a single LLM call to generate several mutated prompts, PW ensures computational efficiency. Figure 5 shows examples of mutated prompts for an initial problem description on the GSM8K.

2. **ScoringComponent:** Next, PW employs a scoring mechanism to evaluate the performance of the generated mutated prompts. The scoring is based on how well each prompt performs against a mini-batch of 5 training examples with ground truth. The scoring mechanism can be either using traditional metrics like F1 score or an LLM as an evaluator, PW supports both. This helps systematically identify the most effective prompt while filtering out underperforming ones. The use of multiple mini-batches ensures robustness in the evaluation. Examples of mutated prompts with their scores are shown in Figure 3 and 5.

3. **CritiqueComponent:** Once the best-performing mutated prompt is selected, PW introduces a unique feedback mechanism through its *critique* component. The critique reviews where the prompt succeeded and failed by analyzing cases where the LLM struggled, such as interpreting

**TOP 3 MUTATED PROMPTS :**
" Let's devise a step-by-step experiment to reach the solution of this mathematical problem."
" Let's simplify and tackle this mathematical problem step by step to make it easier to solve. "
" Let's list out ideas and apply them one by one, thinking step by step, to solve mathematical problems. "

**TOP SCORED PROMPT:**
"Provide question answering on mathematical school grade questions that require multi-step reasoning. The problems should take between 2 and 8 steps to solve, and solutions primarily involve performing a sequence of elementary calculations using basic arithmetic operations (+ - / *) to reach the final answer.\nLets think step by step to arrive at the solution of this mathematical problem",

**CRITIQUE/FEEDBACK :**
Firstly, the instruction doesn't specify the need for the agent to understand the problem context, such as interpreting relationships. Secondly, the instruction lacks clarity on the agent's ability to handle percentages and real-world scenarios. Understanding sequences of actions and their impact are crucial. Lastly, the instruction doesn't mention the agent's ability to handle time conversions, such as converting an hourly rate to a per-minute rate.

**REFINED PROMPT:**
Provide question answering on mathematical school grade problems that require multi-step reasoning and understanding of the problem's context. The problems should take between 2 and 8 steps to solve, and solutions primarily involve performing a sequence of elementary calculations using basic arithmetic operations (+ - / *), handling percentages, and converting them into numbers. The agent should be able to interpret real-world scenarios and understand the implications of the problem, including handling time conversions. The agent should also be able to follow a sequence of actions and their impact on the final answer. Let's think step by step to arrive at the solution of this mathematical problem.

Figure 5: Iterative Prompt Refinement.

**EXAMPLES:**
**[Question]:** Albert is wondering how much pizza he can eat in one day. He buys 2 large pizzas and 2 small pizzas. A large pizza has 16 slices and a small pizza has 8 slices. If he eats it all, how many pieces does he eat that day?
**[Answer]:** He eats 32 from the largest pizzas because 2 x 16 = <<2*16=32>>32. He eats 16 from the small pizza because 2 x 8 = <<2*8=16>>16. He eats 48 pieces because 32 + 16 = <<32+16=48>>48. The answer is 48.

**CRITIQUE/FEEDBACK :**
This example is simple and straightforward, involving basic multiplication and addition. It's relevant and relatable to students. However, it could be improved by adding a bit more complexity, such as introducing fractions or percentages.\n\n

**SYNTHETIC EXAMPLES:**
**[Question]:** Albert buys a pizza that is cut into 8 slices. He eats 3 slices and gives 2 slices to his friend. He then buys another pizza that is cut into 12 slices and eats half of it. How many slices of pizza does Albert have left?
**[Answer]:** From the first pizza, Albert has 8 - 3 - 2 = <<8-3-2=3>>3 slices left. From the second pizza, Albert eats 12 / 2 = <<12/2=6>>6 slices, so he has 12 - 6 = <<12-6=6>>6 slices left.\n\nIn total, Albert has 3 + 6 = <<3+6=9>>9 slices of pizza left. The answer is 9.

Figure 6: Synthetic Example Generation.

**EXPERT IDENTITY :**
You are a mathematics educator with a deep understanding of elementary and middle school mathematics. You are experienced in teaching multi-step problem-solving techniques and have a knack for breaking down complex problems into manageable steps. Your expertise lies in basic arithmetic operations such as addition, subtraction, multiplication, and division. You can provide clear, step-by-step solutions to mathematical problems that require multi-step reasoning. You are patient and thorough, ensuring that each step is clearly explained and understood. Your ability to simplify complex problems and guide students through the problem-solving process makes you an excellent resource for answering school-grade mathematical questions.

**INTENT :** Mathematical Reasoning, Multi-step Problem Solving, Basic Arithmetic Operations, Data Analysis, Solution Verification

Figure 7: Task Intent and Expert Persona.

relationships or time conversions in GSM8k example. This targeted feedback is critical in refining the prompt, as it provides insights into specific weaknesses, allowing for focused improvements rather than general changes. Example of critique feedback on the mutated prompt are in Figure 5.

4. **SynthesizeComponent:** Finally, PW *synthesize* component uses the critique's feedback to refine the best prompt. It rephrases and enhances the instruction based on the critique, producing a more task-specific and optimized prompt. For example, the feedback indicated issues with interpreting specific relationships, the synthesized prompt would address that directly, leading to a clearer, more effective instruction (see Figure 5).

By combining these steps—mutation, scoring, critique, and synthesis—PW ensures that the prompts are not only diverse and creative but also highly tailored to the specific task at hand, outperforming prior methods that lack this guided refinement process.

### 3.2 IDENTIFICATION OF DIVERSE EXAMPLES

Next, we focus on identifying a diverse set of candidate examples to enhance prompt effectiveness. The choice of examples is critical, as diverse representations allow LLMs to better grasp various aspects of the information presented (Rubin et al., 2022; Zhang et al., 2022b; Liu et al., 2022; Chen et al., 2024). We begin by extracting candidate examples from the dataset and employ a scoring mechanism to assess the current prompt's effectiveness against these examples, classifying them into positive and negative categories. Positive examples demonstrate where the prompt succeeds, while negative examples highlight areas for improvement. We randomly select 25 examples and iterate through them to find a targeted number of effective few-shot examples, typically taking five iterations. If this process does not yield the desired count, we randomly select five examples from the initial 25. This targeted approach maximizes efficiency by minimizing the need to evaluate the entire dataset, ensuring that the chosen examples effectively contribute to refining the prompt. The use of both positive and negative examples allows for comprehensive understanding and refinement of prompts.

### 3.3 SEQUENTIAL OPTIMIZATION OF PROMPT INSTRUCTIONS AND FEW-SHOT EXAMPLES

Most existing prompt optimization methods focus on either prompt instructions or few-shot examples. In contrast, PromptWizard (PW) employs a sequential optimization approach that integrates both, enhancing task performance by optimizing them in tandem.

**Few-shot example optimization** follows critique-and-synthesis process: (i) `CritiqueComponent`: PW analyzes previously selected examples, utilizing critique to provide detailed feedback. This feedback is based on error-driven self-reflection, that determines how examples should evolve to be more diverse and task-relevant. (ii) `SynthesizeComponent`: This incorporates feedback from the Critique to generate new synthetic examples that are more diverse, robust, and task-relevant. Figure 6 demonstrates the critique's feedback on a example alongside the newly generated synthetic examples.

**Prompt optimization** follows critique-and-synthesis process: (i) `CritiqueComponent`: The newly generated synthetic examples are evaluated alongside the current prompt. The `CritiqueComponent` identifies weaknesses and gaps that require addressing to further refine the prompt instruction. (ii) `SynthesizeComponent`: This leverages feedback from the critique to synthesize and refine the prompt instruction. This iterative feedback loop facilitates continuous refinement of both the prompt and the synthetic few-shot examples, ensuring they remain aligned with task-specific nuances.

### 3.4 SELF-GENERATED REASONING AND VALIDATION

With the optimized prompt and few-shot examples, we further enhance model performance by incorporating chain-of-thought (CoT) reasoning. Building on the hypothesis that reasoning chains improve problem-solving abilities of the model (Wei et al., 2023; Wang et al., 2023a; Ye et al., 2023). Specifically, we automatically generate a detailed reasoning chain for each selected few-shot examples. (i) `ReasoningComponent`: This takes the selected few-shot examples and generates a detailed reasoning chain for each example to facilitate problem-solving. (ii) `ValidateComponent`: The validation component uses an LLM to check the coherence and relevance of examples (questions,reasoning). This process effectively filters out incorrect examples and/or hallucinated reasoning.

### 3.5 INTEGRATION OF TASK INTENT AND EXPERT PERSONA

To enhance task performance, PW integrates task intent and an expert persona into prompts (Figure 7). (i) `Task Intent`: This ensures that the model stays aligned with task requirements, particularly in specialized domains. By incorporating specific hints or keywords (Sun et al., 2023), derived from the problem description, PW guides the model to apply relevant approaches. We generate these cues using `SynthesizeComponent`, informed by initial problem description. (ii) `Expert Persona`: To maintain consistency and relevance in LLM interactions, we incorporate an expert persona into prompts (Xu et al., 2023). To maintain consistency, PW introduces an expert persona, preventing response variability. This persona is generated based on the problem description and ensures consistent, domain-relevant outputs. All PW components utilize LLMs, with their prompt templates provided in Appendix 16 and algorithmic details in Appendix 14.

## 4 EXPERIMENTS AND IMPLEMENTATION DETAILS

We evaluate `PromptWizard` as a tool to generate instructions and examples that steer a black-box LLM toward desired behavior for a given target task.

**Tasks & Datasets.** We assess the effectiveness of `PromptWizard` on the widely-used BIG-Bench Instruction Induction (BBII) dataset, a benchmark for prompt optimization in recent works such as Instinct (Lin et al., 2024), InstructZero (Chen et al., 2023), and APE (Zhou et al., 2023). The dataset covers a diverse range of language understanding scenarios (Appendix 8).
In addition to BBII, we evaluate `PromptWizard` on three arithmetic reasoning datasets: GSM8k(Cobbe et al., 2021), AQUARAT (Ling et al., 2017), and SVAMP (Patel et al., 2021), as well as domain-specific tasks from BigBench Hard (BBH) (Suzgun et al., 2022), which includes 23 challenging tasks. This brings the total to 45 tasks (19 BBII, 23 BBH, 3 math tasks), covering both general and domain-specific problem settings. Additional details of all datasets are in Appendix 9.

**Baselines.** We compare our `PromptWizard` with five representative SOTA discrete and continuous methods: **Instinct** (Lin et al., 2024), **InstructZero** (Chen et al., 2023), **PromptBreeder** (PB) (Fernando et al., 2023), **EvoPrompt** (Guo et al., 2024), and **APE** (Zhou et al., 2023).

**Implementation Details.** We experiment with both ChatGPT (`GPT3.5Turbo`) and `GPT-4` as the black-box LLMs for prompt optimization in `PromptWizard`. All the individual components such as mutate, score, critique, reason, synthesize and validate, rely on the same LLM either `GPT3.5Turbo` or `GPT-4`, accordingly. For all experiments, we use only 25 examples from the training data to optimize the prompts and in-context examples, with evaluations conducted on the full test dataset. To ensure robustness, all reported results are averaged over three experimental runs. Details of the hyperparameters used in the paper are provided in Appendix 10. Specifically, we restrict the number of mutated prompts & mutation rounds to 3, diverse examples to 25, sequential optimization rounds to 5. The anonymized source code of `PromptWizard` is available for reproducibility[1].

---

[1]Anonymized source code: `https://anonymous.4open.science/r/PromptWizard/`

Table 1: Average test accuracy achieved by best instruction generated by different SOTA algorithms. InsZero: InstructZero, PB: PromptBreeder, EvoP: EvoPrompt, PW: PromptWizard (ours).

| Task | APE | InsZero | PB | EvoP | Instinct | PW | Instinct | PW |
|---|---|---|---|---|---|---|---|---|
| **LLM: GPT3.5Turbo** | | | Zero-shot setting | | | | One-shot setting | |
| antonyms | 0.64 | 0.83 | 0.80 | 0.80 | **0.85** | 0.56 | **0.85** | 0.78 |
| auto-categorization | 0.25 | 0.26 | 0.22 | 0.26 | 0.25 | **0.28** | 0.30 | **0.40** |
| cause and effect | 0.57 | 0.81 | 0.75 | 0.83 | 0.59 | **0.88** | 0.63 | **0.92** |
| common concept | 0.07 | 0.09 | 0.10 | 0.12 | **0.21** | 0.10 | **0.25** | 0.19 |
| diff | 0.67 | 0.69 | **1.00** | **1.00** | **1.00** | **1.00** | **1.00** | **1.00** |
| informal to formal | 0.57 | 0.53 | 0.58 | **0.62** | 0.55 | **0.62** | 0.52 | **0.56** |
| letters list | **1.00** | 0.59 | 0.99 | **1.00** | **1.00** | 0.95 | **1.00** | **1.00** |
| negation | 0.75 | 0.78 | 0.77 | 0.79 | **0.82** | 0.73 | **0.86** | 0.84 |
| object counting | 0.36 | 0.36 | 0.34 | 0.12 | 0.34 | **0.60** | 0.36 | **0.52** |
| odd one out | 0.63 | 0.61 | 0.64 | 0.65 | 0.70 | **0.78** | 0.63 | **0.92** |
| orthography starts with | 0.46 | 0.51 | 0.56 | 0.60 | 0.67 | **0.75** | 0.67 | **0.92** |
| rhymes | 0.16 | **1.00** | 0.54 | 0.61 | **1.00** | 0.89 | 0.75 | **0.90** |
| second word letter | 0.75 | 0.43 | 0.57 | 0.41 | 0.10 | **0.93** | 0.24 | **0.99** |
| sentence similarity | 0.00 | 0.00 | 0.01 | 0.28 | 0.14 | **0.29** | 0.16 | **0.30** |
| sum | 0.67 | **1.00** | **1.00** | **1.00** | **1.00** | **1.00** | **1.00** | **1.00** |
| synonyms | 0.36 | 0.28 | 0.36 | 0.14 | 0.31 | **0.37** | 0.37 | **0.44** |
| taxonomy animal | 0.35 | 0.72 | 0.72 | 0.72 | 0.86 | **0.92** | 0.90 | **0.94** |
| word sorting | 0.33 | 0.31 | **0.56** | 0.52 | 0.51 | **0.56** | 0.62 | **0.74** |
| word unscrambling | 0.44 | 0.55 | 0.61 | 0.60 | **0.63** | 0.52 | **0.58** | **0.58** |
| #best performing tasks | 1 | 2 | 3 | 4 | 8 | **13** | 7 | **16** |

# 5 EXPERIMENTAL RESULTS AND ANALYSIS

## 5.1 PERFORMANCE ANALYSIS AGAINST VARIOUS PROMPTING BASELINES

**Zero-shot accuracy.** We evaluate the zero-shot test accuracy of ChatGPT (GPT3.5Turbo) using instructions generated by five methods: APE, InstructZero, PromptBreeder, EvoPrompt, and Instinct. Table 1 presents results on 19 challenging tasks from BIG-Bench Instruction Induction (BBII) dataset, selected where the average test accuracy across all methods is below 0.8, following the evaluation protocol in Instinct (Lin et al., 2024). All experiments use the same black-box LLM (GPT3.5Turbo) under a zero-shot setting, ensuring a fair and consistent comparison across methods. PromptWizard outperforms the baselines, achieving the highest accuracy on 13 out of 19 tasks (68%), compared to Instinct's 8 tasks (42%). This significant improvement demonstrates PromptWizard's strength in tackling complex instruction induction tasks.

**Overall Performance.** Figure 2 shows the performance profile curve for the instruction induction tasks from Table 1. The performance profile curve (Dolan & Moré, 2002) visualizes how frequently different approaches' performance is within a given distance of the best performance. In this curve, the x-axis ($\tau$) represents the performance ratio relative to the best-performing method, and the y-axis ($p(\tau)$) reflects the fraction of tasks where a method's performance is within this ratio. So for a given method, the curve tells what percentage of the tasks are within $\tau$ distance to the best performance (among different methods). PromptWizard consistently outperforms other methods across various thresholds, maintaining the highest $p(\tau)$ values, indicating that it consistently performs near the best possible accuracy across all tasks. Additional analysis is available in Appendix 11.

**One-shot Accuracy.** To evaluate the effectiveness of PW's in-context example generation, we compare the one-shot test accuracy of ChatGPT (GPT3.5Turbo) when using instructions generated by Instinct and PW. The results, presented in the last two columns of Table 1, show that PromptWizard achieves the highest accuracy on 16 out of 19 tasks (84%), while Instinct performs best on only 7 out of 19 tasks (36%). This improvement is largely attributed to the robust in-context learning examples generated by PW, combined with its iterative prompt instruction optimization. By refining both the prompt instructions and examples through multiple iterations, PW ensures that the task-specific knowledge is effectively captured. The optimal prompts are in Appendix 15.

**GPT-4 as Base model.** Table 1 presents results using GPT3.5Turbo as the base model. In additional experiments with GPT-4 as the base model on BBII, PW achieved the highest accuracy in 15 out of 19 tasks (79%), compared to Instinct's 6 out of 19 (31%), demonstrating PW's superior performance even with a change in base models (Appendix 12 Table 12 has the detailed results).

**Arithmetic Datasets.** Table 2 compares performance of PW with Instinct and InstructZero on three arithmetic reasoning tasks: GSM8k, AQUARAT, and SVAMP, all using GPT3.5Turbo in a zero-shot setting. The results clearly show that PromptWizard consistently outperforms all

Table 2: Perf. on arithmetic tasks.

| Dataset | GSM8k | AQUARAT | SVAMP |
|---|---|---|---|
| Approach | Zero-shot with `GPT3.5Turbo` | | |
| InsZero | 74.2 | 54.3 | 79.5 |
| Instinct | 74.5 | 54.7 | 81 |
| PW | **90** | **58.2** | **82.3** |

Table 3: Perf. on BBH.

| Dataset | BBH (23) |
|---|---|
| Approach | Accuracy |
| APE | 71.85 |
| EvoP | 75.03 |
| PW | **88.1** |

Table 4: Cost analysis.

| | API calls | IO Tokens | Total tokens | Cost ($) |
|---|---|---|---|---|
| Instinct InsZero | 1730 | 67 | 115910 | 0.23 |
| PB | 18600 | 80 | 1488000 | 2.9 |
| EvoP | 5000 | 80 | 400000 | 0.8 |
| PW | 69 | 362 | 24978 | 0.05 |

baselines across these datasets, achieving significant gains in accuracy on arithmetic reasoning tasks. These tasks, often requiring detailed multi-step reasoning, which PW addresses through its iterative synthesis of prompts enriched with intermediate reasoning steps and examples.

**Comparison with BBH tasks.** In Table 3, we report the average accuracy across 23 tasks from the BIG-Bench Hard (BBH) dataset. Due to the high cost and compute requirements involved in evaluating all baselines on this extensive set of tasks, we limit the comparison to EvoPrompt and APE. `PromptWizard` achieves a remarkable improvement, increasing the average accuracy by over 13% compared to EvoPrompt and APE, underscoring its effectiveness in handling complex tasks.

## 5.2 COST ANALYSIS AGAINST VARIOUS PROMPTING BASELINES

While high accuracy is crucial, the efficiency of generating prompts is equally important. We present a detailed cost analysis demonstrating that PW not only outperforms baselines in terms of task accuracy but does so with minimal computational overhead. We conduct a comprehensive evaluation by computing the total number of API calls, tokens processed, and the corresponding cost (Table 4).

**Instinct and InstructZero.** Instinct and InstructZero use a mix of white-box and black-box models to continuously optimize soft prompts, with the number of API calls linked to the iterative process needed for convergence. According to their respective papers, the best performance is typically achieved after a maximum of 165 iterations. On average, across all tasks, we observed **1,730 API calls** to the black-box model per task, with approximately 67 input and output (IO) tokens per call for the BBII dataset. Given the token billing structure of the `GPT3.5Turbo` API ($0.002 per 1,000 tokens), the total cost per task is estimated to be around **$0.23**. Detailed API call and token breakdowns per task are provided in Appendix 13.2.

**PromptBreeder (PB).** PromptBreeder (PB) uses a discrete optimization approach through self-referential improvement, evolving prompts over 20–30 generations with a population size of 20. This results in significant API usage, with an estimated **18,600 API calls** per task (30 generations × (20 mutations + 20×30 evaluations)) (Fernando et al., 2023). With an average of 80 input/output tokens per call, the total cost per task for the BBII dataset is approximately **$2.9**, making PB one of the most expensive methods among the baselines.

**EvoPrompt.** EvoPrompt, a discrete optimization method, uses evolutionary algorithms to find optimal prompts. The number of API calls follows the formula: API calls = N (population size) × T (iterations) × (1 + D (development size)). For BBII tasks, with a population size of 10, 10 iterations, and a development set size of 50, this results in: API calls = 10×10×(1+50) = **5,000 API calls**. With an average of 80 input/output tokens per call, EvoPrompt incurs a total cost of **$0.8 per task**, which is lower than PB but still considerable compared to other methods.

**`PromptWizard` (PW).** PW employs a discrete optimization, similar to PB and EvoPrompt, but introduces key components- feedback-driven guided exploration, critique and synthesis process, and sequential optimization of instruction and examples- that streamline prompt exploration and focus on meaningful evolution. These innovations reduce unnecessary mutations, striking an effective balance between exploration and exploitation. The API calls in PW are broken down into 48 for prompt refinement, 5 for example selection, 12 for sequential optimizations, and 4 for reasoning, validation, intent refinement, and expert identity (Algo. 1). This totals **69 API calls**, substantially fewer than PB's 18,600 and EvoPrompt's 5,000. The average input/output tokens per task is around 360, slightly higher due to the addition of COT reasoning and expert identity during prompt optimization. Despite this, `PromptWizard` costs **just $0.05 per task** with 5-60x reduction in overall tokens, significantly lower than other techniques. Note that, during inference, PW's average input tokens are ∼200, which is comparable to other approaches. Appendix 13.2 shows the detailed task level computations.

*`PromptWizard`'s efficiency is highlighted by being **5x cheaper** than continuous methods like Instinct and InstructZero, and **16-60x cheaper** compared to discrete methods like EvoPrompt and PromptBreeder, while achieving superior performance.*

| Datasets | 5 (eg) | 25 (eg) |
|----------|--------|---------|
| MMLU     | 80.4   | 89.5    |
| GSM8k    | 94.0   | 95.4    |
| Ethos    | 86.4   | 89.4    |
| PubMedQA | 68.0   | 78.2    |
| MedQA    | 80.4   | 82.9    |
| Average  | **81.9** | **87.0** |

Table 5: Perf. with 5 examples.

| Datasets | Ll-70B | GPT-4 |
|----------|--------|-------|
| GSM8k    | 94.6   | 95.4  |
| Ethos    | 89.2   | 89.4  |
| Average  | **91.9** | **92.4** |

Table 6: Perf. with smaller LLM for prompt generation. Ll-70B: `Llama-70B`

| Models | With PW | w/o PW |
|--------|---------|--------|
| `GPT-4` | **95.4** | 92   |
| GPT3.5 | **75.6** | 57.1   |
| Ll-70B | **90.2** | 56.8   |

Table 7: Perf. with different Base LLMs on GSM8k. Ll-70B: `Llama-70B`

## 6 PROMPTWIZARD ABLATION STUDY

### 6.1 PROMPTWIZARD EFFICACY WITH FEWER TRAINING EXAMPLES

`PromptWizard` assesses prompt effectiveness using available training examples while also synthesizing new few-shot examples. In real-world scenarios, where data may be scarce or tasks evolve without curated datasets, generating effective prompts with minimal examples becomes essential. To evaluate `PromptWizard`'s performance under data-constrained conditions, we simulate a few-shot learning scenario by randomly selecting only *5* examples from each dataset as the training set (instead of 25). PW utilizes these examples for all evaluations, critique feedback, and the generation of diverse synthetic examples. This setup tests the framework's ability to generalize and create robust, task-relevant prompts with minimal data.

Table 5 showcases `PromptWizard`'s performance across five diverse datasets (see Appendix 9) when trained with only 5 examples (**5 eg**) compared to 25 examples (**25 eg**). Despite the drastic reduction in training data, `PromptWizard` demonstrates impressive resilience, exhibiting only a marginal **5% drop in accuracy** on average. This resilience underscores the model's adaptability, driven by two key mechanisms: (i) *Synthetic Example Generation* using critique-and-synthesize, which produces diverse, high-quality examples from limited inputs, reducing the impact of data scarcity; and (ii) *Reasoning Chain Guidance*, where structured reasoning chains enhance the LLM's ability to generate accurate, contextually relevant responses.

### 6.2 PROMPTWIZARD WITH SMALLER LLMS FOR PROMPT OPTIMIZATION

In prior experiments, `GPT3.5Turbo` was used for both prompt generation and optimization. In this section, we explore the feasibility of employing a smaller LLM, such as `Llama-70B`, for prompt generation while reserving a more capable model like `GPT-4` for inference. This approach reduce computational costs during prompt optimization by leveraging the efficiency of smaller models while still maximizing task accuracy with powerful model during inference. This strategy offers two key advantages: (i) *Computational Efficiency*: Smaller LLMs like `Llama-70B` require fewer resources, making them ideal for generating prompts in resource-constrained environments. (ii) *Task Performance*: Despite using a smaller model for prompt generation, inference benefits from the larger `GPT-4` model's ability to interpret and execute the optimized prompt, ensuring minimal degradation.

Table 6 compares task accuracy across multiple datasets when `Llama-70B` is used for prompt generation versus the default `GPT-4`. Impressively, the final prompts generated by `PromptWizard` using `Llama-70B` show a negligible **<1% drop in accuracy** compared to those generated with `GPT-4`, highlighting `PromptWizard`'s effectiveness even with smaller models. While we experimented with smaller models like Llama-3-8B, they struggled to generate complex instructions, leading to significant performance degradation. Thus, mid-sized LLMs like `Llama-70B` are recommended for prompt optimization, striking a balance between computational efficiency and task performance. These findings demonstrate `PromptWizard`'s adaptability and its ability to maintain high performance across different model sizes with minimal loss.

### 6.3 ASSESSING PERFORMANCE WITH DIFFERENT BASE LLMS

We perform two types of ablation analysis: (i) evaluating the effect of different base LLMs during prompt optimization and inference, and (ii) measuring the contribution of each component within the `PromptWizard` framework to overall performance.

**Ablation on Different Base LLMs.** To assess `PromptWizard`'s adaptability and efficacy across various LLMs, we experiment with three settings: using `GPT-4`, `GPT3.5Turbo`, and `Llama-70B` as both the base LLM for prompt optimization and during inference. The goal is to understand whether the choice of base model impacts the performance gains achieved through `PromptWizard`.

Table 7 summarizes the results for the GSM8k dataset. In case of without PW, we use few-shot learning with Chain-of-Thought (COT) prompting (Touvron et al., 2023) as the baseline. We observe substantial performance improvements across all models when optimized prompts are generated by PW. Specifically, for `GPT3.5Turbo`, the task accuracy increases by +18%, while for `Llama-70B`, the improvement is even more pronounced, reaching +33%. In contrast, models when not using PW prompt show significant performance degradation, reaffirming the value of prompt optimization.

**Effectiveness of different stages of `PromptWizard`.** We conducted an ablation study to assess the contribution of each stage in the PW pipeline, using the GSM8k and Ethos datasets.

|  | GSM8k | Ethos |
|---|---|---|
| All | **95.4** | **89.4** |
| No Mutation and Scoring | 95.2 | 87.1 |
| No Critique and Synthesize | 90.9 | 86.9 |
| No intent & Expert | 95 | 88.7 |
| No Reasoning | 45.9 | 87.6 |

Table 8: Abaltion Study

Table 8 presents the results of this ablation study: (i) *Mutation and Scoring:* The initial stage of iterative prompt refinement alone yields an accuracy boost of 1-2%, demonstrating the baseline value of exploring prompt variations. (ii) *Critique Feedback and Refinement:* Adding structured feedback via the critique mechanism improves accuracy by 3-5 highlighting the impact of targeted refinement on prompt quality. (iii) *Task Intent and Expert Persona Modeling:* Tailoring prompts to task-specific nuances contributes an additional 0.5-1% improvement. Although smaller, this step plays a crucial role in aligning the prompt with task-specific behavior. (iv) *Reasoning on Few-shot Examples:* This emerges as one of the most significant contributors, indicating that generating detailed reasoning chains for few-shot examples is critical for task accuracy. This ablation study underscores the significance of individual components within the `PromptWizard`, as they work collectively to enhance prompt and model performance.

# 7 CONCLUSIONS

This work introduces `PromptWizard`, a general-purpose framework for automating prompt and example synthesis. By striking a balance between exploration and exploitation through a feedback-driven critique and synthesis process, PW systematically refines prompts and in-context examples to enhance task performance. Extensive evaluations across diverse datasets show it consistently outperforms state-of-the-art methods, demonstrating strong efficacy even with limited training data and smaller LLMs, with only a marginal drop in accuracy. Ablation studies highlight the importance of each stage in refining prompts, generating diverse examples, and improving reasoning. Our comprehensive cost analysis highlights significant reductions in API calls, token usage, and overall expenses, showcasing PW's cost-effectiveness—it is 5x cheaper than continuous optimization methods and 16-60x cheaper than discrete methods, all while delivering superior performance. This work democratizes access to effective prompt engineering, enabling more efficient and accurate utilization of LLMs across various domains and applications. Future work will focus on refining the validation of synthetic examples and applying PW to real-world, resource-constrained environments.

**Limitations:** While we have conducted extensive experiments across a diverse set of tasks, careful validation is required for new tasks to ensure adaptability. Prompt response testing is essential before real-world deployment to verify effectiveness. Additionally, while PromptWizard automates prompt engineering, human expertise remains indispensable in guiding and refining the optimization process.

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

APPENDIX

## 8 BIG BENCH INSTRUCTION INDUCTION (BBII) DATASET DETAILS

Table 9 describes the numerous tasks in BBII dataset along with the description of the task. This is a popular dataset and the selected tasks cover many facets of language understanding and includes all nine such problems from the BigBench-Hard Subset. In particular, it includes emotional understanding, context-free question answering, reading comprehension, summarization, algorithms, and various reasoning tasks (e.g., arithmetic, commonsense, symbolic, and other logical reasoning tasks). We selected tasks for which the data was publicly available.

Table 9: Big Bench Instruction Induction Dataset

| Task | Description |
|---|---|
| antonyms | Make the pairs of words opposite. |
| auto categorization | Create a list of things that the input could be associated with, and the output would be the category that the input belongs to |
| cause and effect | identify the sentence that is the cause of the effect in the input sentence pair |
| common concept | "involve" the objects mentioned in the input, so the answer would be "involve oscillations" for the input "guitars, pendulums" |
| diff | Find the difference between the two numbers |
| informal to formal | convert the input sentence into an output sentence that is grammatically correct and idiomatic in English |
| letters list | output the input with a space after each letter |
| negation | make the output false by adding the word "not" to the input |
| object counting | output the number of objects in the input list |
| odd one out | find the word that is most dissimilar to the others in the group |
| orthography starts with | output the word that starts with the letter that was inputted |
| rhymes | output the first word that appeared in the input text |
| second word letter | takes a string as input and returns the first character that is a vowel. |
| sentence similarity | Find the difference between the two sentences and the output was 4 - almost perfectly |
| sum | add the numbers of the two input numbers |
| synonyms | create a list of words that could be used in the same way as the original words |
| taxonomy animal | output the name of an animal that starts with the letter |
| word sorting | sort the input words alphabetically |
| word unscrambling | output the word that is formed by rearranging the letters of the given word |

## 9 DATASET DETAILS: TRAIN/TEST SPLIT FOR DATASETS & FEW-SHOT COUNT

Below are the details of the datasets used for evaluation.

| Datasets | Test dataset size | Few-shot count |
|---|---|---|
| GSM8k | 1319 | 5 |
| AQUARAT | 254 | 0 |
| SVAMP | 254 | 0 |
| Ethos | 799 | 3 |
| PubMedQA | 500 | 5 |
| MedQA | 1273 | 5 |
| CSQA | 1140 | 5 |
| SQA | 224 | 5 |
| BBH ['snarks', 'penguins in a table', 'causal judgement'] | 153, 121, 162 | 3 |
| BBH all except ['snarks', 'penguins in a table', 'causal judgement'] | 225 | 3 |
| MMLU [clinical knowledge, college biology, college medicine, anatomy, medical genetics, professional medicine] | 65, 144, 173, 135, 100, 272 | 5 |

Table 10: Train/Test split for datasets & Few-shot count

`GSM8K`: This dataset contains 8.5K high-quality, linguistically diverse grade school math word problems created by human problem writers. The final answer is an integer value.

`AQUARAT`: A large-scale dataset consisting of approximately 100,000 algebraic word problems. The solution to each question is explained step-by-step using natural language. The test data includes 254 questions.

`SVAMP`: SVAMP (Simple Variations on Arithmetic Math word Problems) dataset is a one-unknown arithmetic word problems with grade level up to 4 by applying simple variations over word problems in an existing dataset.

`Ethos`: This hate speech detection dataset is built from YouTube and Reddit comments. It includes two tasks: binary classification and multi-label classification. We evaluate our approach on the binary classification task, which consists of 998 questions. The final answer is either "yes" or "no."

`MedQA`: This dataset includes multiple-choice questions similar to those in the Medical Licensing Examination. We use the English subset with 11,450 training and 1,273 test questions, styled like the United States Medical Licensing Exam (USMLE). The final answer is the correct option from the available choices.

`MMLU`: Measuring Massive Multitask Language Understanding (MMLU) includes multiple-choice exam questions from 57 domains. We use 6 medical datasets, *viz.,* Clinical knowledge, Medical genetics, Anatomy, Professional Medicine, College Biology, and College Medicine.

`BBH`: BIG-Bench Hard (BBH) includes 23 tasks from different domains. Answers can be in the form of multiple-choice questions, boolean, or string responses.

For all the datasets, in `PromptWizard` we randomly select only 25 samples from available training data. We do not use entire training dataset in training-phase. Test dataset size for each dataset is specified below. However for the baseline approaches, we follow their train/test splits. Table 10 provides details of the test set along with the few-shots used in each dataset.

## 10 HYPER PARAMETERS

PW relies on several parameters to control the level of exploration and evolution at each stage. We now provide comprehensive details of all parameters and associated values (see Table 11).

## 11 PERFORMANCE PROFILE CURVE - ADDITIONAL DETAILS

In Section 5.1 we presented the Performance Profile Curve comparing `PromptWizard`'s performance against all baselines across all tasks in BBII dataset.

The performance profile curve Dolan & Moré (2002) visualizes how frequently different approaches' performance is within a given distance of the best performance. In this curve, the x-axis ($\tau$) represents the performance ratio relative to the best-performing method, and the y-axis ($p(\tau)$) reflects the fraction of tasks where a method's performance is within this ratio. `PromptWizard` consistently

| Hyper-parameter | Description | Default Value |
|---|---|---|
| *mutate_refine rounds* | Number of rounds of call to `MutateComponent` followed by refinement over best prompt among generated by `MutateComponent` in previous step. | 3 |
| *mutate_rounds* | Number of times `MutateComponent` would be called. | 3 |
| *style_variation* | Number of variations `MutateComponent` generates in a single call. i.e. one variation corresponding to each thinking style provided. | 3 |
| *min_example correct_count* | Minimum number of questions the `ScoringComponent` should answer correctly for a prompt to get qualified for next stage. | 3 |
| *max_example count* | Maximum number of attempts/questions the `ScoringComponent` would be asked asked to answer. | 6 |
| *max_seq_iter* | Number of rounds of call to `CritiqueComponent` followed by call to `SynthesizeComponent` | 5 |
| *few_shot_count* | Total number of few shot examples to be provided in prompt. | Defined in Table 10 |
| *ex_critique* | Number of LLM calls made by `CritiqueComponent` for getting critique for improving examples passed as few-shots. | 1 |
| *synthesize* | Number of LLM calls made by `SynthesizeComponent` to generate synthetic examples. | 1 |
| *inst_critique* | Number of LLM calls made by `CritiqueComponent` for getting critique for improving instruction passed as few-shots. | 1 |
| *synthesize* | Number of LLM calls made by `SynthesizeComponent` to created improved version of instruction. | 1 |
| *reasoning + validation* | Number of LLM calls made by `ReasoningComponent` and `ValidateComponent` respectively. | 2 |
| *intent + persona* | Number of LLM calls made to get keywords that express the intent and to generate expert persona respectively. | 2 |

Table 11: Description for hyper parameters and their default values

outperforms other methods across various thresholds, maintaining the highest $p(\tau)$ values, indicating that it consistently performs near the best possible accuracy across all tasks.

In this curve, the x-axis ($\tau$) represents the performance ratio relative to the best-performing method, and the y-axis ($p(\tau)$) reflects the fraction of tasks where a method's performance is within this ratio. It is a suitable measure for the performance of methods over a large number of tasks. To draw the performance profile curve for a method, for each task $i$, we check whether the performance of this method in task i is within $\tau$ distance to the best performance (among different methods) in task $i$, and define an indicator function $I()$. Next, we average this indicator function across all $n_p$ tasks, which yields a value $p(\tau)$ (equation 1). Finally, the performance profile curve for this method is obtained by varying the value of $\tau$ and calculating the corresponding $p(\tau)$.

$$\rho(\tau) = \frac{\sum_{i=1}^{n_p} \mathbb{I}\left(\text{Best performance of task } i - \text{Performance of the approach on task } i \le \tau\right)}{n_p} \quad (1)$$

For example at $\tau = 0.0$, the values of $p(\tau)$ are approximately 0.05 (APE), 0.105 (InstructZero), 0.157 (PromptBreeder), 0.210 (EvoPrompt), 0.421 (INSTINCT), 0.68 (`PromptWizard`). This shows that `PromptWizard` is the best performing method, betting all the other methods at 68% of the tasks.

Table 12: Average test accuracy achieved by best instruction generated by Instinct and PW using GPT4 as base model on BBII dataset.

| Task | Instinct | PromptWizard |
|------|----------|--------------|
| **LLM: GPT4** | Zero-shot setting | |
| antonyms | **0.79** | 0.77 |
| auto categorization | 0.3 | **0.38** |
| cause and effect | **0.96** | 0.88 |
| common concept | **0.2** | 0.15 |
| diff | **1** | **1** |
| informal to formal | 0.6 | **0.75** |
| letters list | **1** | **1** |
| negation | 0.7 | **0.85** |
| object counting | 0.6 | **0.82** |
| odd one out | 0.54 | **0.87** |
| orthography starts with | 0.75 | **0.92** |
| rhymes | **1** | 0.88 |
| second word letter | 0.57 | **0.97** |
| sentence similarity | 0.3 | **0.43** |
| sum | 0.99 | **1** |
| synonyms | 0.3 | **0.42** |
| taxonomy animal | 0.9 | **1** |
| word sorting | 0.5 | **0.65** |
| word unscrambling | 0.54 | **0.77** |
| # best performing tasks | 6 | **15** |

---

**Algorithm 1** Total LLM Calls Calculation

---

1: **Calculation**: Input: Hyperparameters, Result: Total LLM Calls
2: **refine_instructions_component** $\leftarrow$ mutate_refine_rounds $\times$ (mutate_rounds $\times$ style_variations + min_example_correct_count + critique + synthesize)
3: **seq_iter_component** $\leftarrow$ max_seq_iter $\times$ (ex_critique + ex_synthesize + inst_critique + inst_synthesize)
4: **other_components** $\leftarrow$ max_example_count + reasoning + validation + intent + persona
5: **Total LLM Calls** $\leftarrow$ refine_instructions_component + seq_iter_component + other_components
6: Total LLM calls $= \{3 \times ((3 \times 3) + 5 + 1 + 1)\} + \{5\} + \{3 \times ((1 + 1) + (1 + 1))\} + \{1 + 1\} + \{1 + 1\} = 48 + 5 + 12 + 2 + 2$
7: Prompt_refinement = 48; example_selection = 5; seq_opt = 12;
8: reason+validate = 2; intent+expert = 2
9: Total LLM calls $= 69$

---

## 12 ADDITIONAL RESULTS: BBII DATASET

Table 12 shows additional experiments with `GPT-4` as the base model, PW achieved the highest accuracy in 15 out of 19 tasks, compared to Instinct's 6 out of 19, demonstrating PW's superior performance even with a change in base models.

## 13 COST ANALYSIS: ADDITIONAL DETAILS

### 13.1 PROMPTWIZARD LLM API CALLS CALCULATION

We compute the total LLM calls made by `PromptWizard` during prompt generation (one-time), which derives the most effective prompt and few-shot examples. The algorithm provides more details: Algorithm 1 describes the total LLM calls made by `PromptWizard` during preprocessing (one-time), which derives the most effective prompt and few-shot examples (see Appendix 10.for parameter description). Note that during inference, each query uses only the default *one* LLM call.

Table 13: Cost analysis of Instinct and PromptWizard on BBII dataset with `GPT3.5Turbo` as the base model.

| Dataset | Instinct | | PromptWizard | |
|---|---|---|---|---|
| | API Calls | IO Tokens | API Calls | IO Tokens |
| antonyms | 2200 | 39 | 69 | 334 |
| auto-categorization | 1740 | 86 | 69 | 341 |
| cause and effect | 1352 | 61 | 69 | 390 |
| common concept | 639 | 94 | 69 | 386 |
| diff | 1820 | 58 | 69 | 381 |
| informal to formal | 880 | 90 | 69 | 271 |
| letters list | 2240 | 58 | 69 | 256 |
| negation | 2180 | 60 | 69 | 305 |
| object counting | 1340 | 69 | 69 | 470 |
| odd one out | 840 | 50 | 69 | 372 |
| orthography starts with | 1800 | 82 | 69 | 339 |
| rhymes | 1920 | 41 | 69 | 391 |
| second word letter | 1840 | 48 | 69 | 257 |
| sentence similarity | 2140 | 78 | 69 | 626 |
| sum | 2180 | 66 | 69 | 367 |
| synonyms | 2100 | 51 | 69 | 452 |
| taxonomy animal | 1900 | 72 | 69 | 225 |
| word sorting | 1680 | 110 | 69 | 426 |
| word unscrambling | 2060 | 58 | 69 | 306 |
| Average | 1729 | 67 | 69 | 362 |

## 13.2 COMPARISON OF API CALLS, NUMBER OF TOKENS FOR BBII DATASET

Table 13 shows the comparison of API calls, number of tokens for BBII dataset for both Instinct and `PromptWizard` using `GPT3.5Turbo` model. We can see that PW has significant lower number of API calls compared to Instinct, thus resulting in 5x reduction in overall tokens per task. Similar trends with the API calls, number of tokens used, were seen when the base model in Instinct and PW was changed to `GPT-4`.

## 14 PROMPTWIZARD ALGORITHM

Algorithm 2 provides pseudo code for entire `PromptWizard` framework. Algorithm 3 provides pseudo code for mutating prompt instruction and further refining the best prompt instruction among all the mutated prompt instructions. i.e. Section 3.1. Algorithm 4 and 5 provide pseudo code for Sections 3.2 and 3.3 respectively.

---

**Algorithm 2** `PromptWizard` Framework

---

1: **Input:** $L$: large language model; $D$: problem description; $S$: set of training samples $\{(q_i, a_i)\}_{i=1}^N$; $T$: thinking styles; $N$: *mutate_refine_rounds*; $k$: few-shot count ; $N_1$: *max_seq_iter*

2: **Output:** Optimized prompt $\hat{P}_{\text{opt}}$ and few-shot examples $\{(q_{f_i}, a_{f_i})\}_{i=1}^k$

3: **procedure** PROMPTWIZARD($L$, $D$, $S$, $T$, $k$, $N$, $N_1$)

4:      Initialize $P \leftarrow$ initial prompt instruction

5:      $\hat{P} \leftarrow$ RefineInstructions($L$, $D$, $S$, $T$, $N$)

6:      $\mathcal{E}_{\text{diverse}} = \{(q_{d_i}, a_{d_i})\}_{i=1}^k \leftarrow$ DiverseExampleSelection($L$, $D$, $S$, $\hat{P}$)

7:      $\hat{P}_{\text{opt}}, \mathcal{E}_{\text{syn}} = \{(q_{s_i}, a_{s_i})\}_{i=1}^k \leftarrow$ SequentialOptimization($L$, $\hat{P}$, $\mathcal{E}_{\text{diverse}}$, $N_1$)

8:      $\mathcal{E}_{\text{syn,r}} \leftarrow$ `ReasoningComponent` ($\mathcal{E}_{\text{syn}}$)      ▷ generate reasoning chains

9:      $\{(q_{f_i}, a_{f_i})\}_{i=1}^k \leftarrow$ `ValidateComponent` ($\mathcal{E}_{\text{syn,r}}$)      ▷ validate examples

10:      $\tau_{\text{intent}} \leftarrow$ `SynthesizeComponent` ($D$)      ▷ generate task intent

11:      $\pi_{\text{expert}} \leftarrow$ `SynthesizeComponent` ($D$)      ▷ generate expert persona

12:      **return** $\pi_{\text{expert}}, \hat{P}_{\text{opt}}, \{(q_{f_i}, a_{f_i})\}_{i=1}^k, \tau_{\text{intent}}$

13: **end procedure**

---

---

**Algorithm 3** RefineInstructions Procedure

---

1: **Input:** $L$: large language model; $D$: problem description; $S$: set of training samples $\{(q_i, a_i)\}_{i=1}^N$; $T$: thinking styles; $N$: *mutate_refine_rounds*; $b$: batch size (default: 5); $v$: number of thinking styles to select; $M$: *mutate_rounds*

2: **Output:** Optimized prompt $\hat{P}$

3: **procedure** REFINEINSTRUCTIONS($L$, $D$, $S$, $T$, $N$, $b$, $v$, $M$)

4:  Initialize $P \leftarrow$ initial prompt instruction

5:  Optimized prompt $\hat{P} \leftarrow P$

6:  **for** *refinement_round* $= 1$ to $N$ **do**

7:    $T_1 \leftarrow$ RandomlySelect($v, T$)        ▷ Select $v$ thinking styles from $T$

8:    $\mathcal{F} \leftarrow \emptyset$

9:    **for** $m = 1$ to $M$ **do**

10:     $\mathcal{M} \leftarrow$ MutateComponent($D$,P,$T_1$)

11:     **for** $p \in \mathcal{M}$ **do**

12:      $s \leftarrow$ ScoringComponent($p, S, b$)

13:      **if** $s > 0.5$ **then**

14:       $\mathcal{F} \leftarrow \mathcal{F} \cup \{(p, s)\}$

15:      **end if**

16:     **end for**

17:    **end for**

18:    *top_scored_prompt* $\leftarrow \arg\max_{p \in \mathcal{F}}\{s(p)\}$

19:    *feedback* $\leftarrow$ CritiqueComponent(*top_scored_prompt*)

20:    $\hat{P} \leftarrow$ SynthesizeComponent(*top_scored_prompt*,*feedback*)

21:  **end for**

22:  **return** $\hat{P}$

23: **end procedure**

---

---

**Algorithm 4** DiverseExampleSelection Procedure

---

1: **Input:** $L$: large language model; $D$: problem description; $S$: training dataset $\{(q_i, a_i)\}_{i=1}^N$; $k$: few-shot count

2: **Output:** Selected diverse examples $\mathcal{E}_{\text{diverse}} = \{(q_{d_i}, a_{d_i})\}_{i=1}^k$

3: **procedure** DIVERSEEXAMPLESELECTION($L$, $D$, $S$, $k$)

4:  $S' \leftarrow$ RandomSample($S, 25$)

5:  $\mathcal{E}_{\text{diverse}} \leftarrow \emptyset$

6:  count $\leftarrow 0$

7:  **for** $(q, a) \in S'$ **do**

8:   $a_{\text{pred}} \leftarrow L(q)$                ▷ LLM's answer for $q$

9:

10:   **if** $a_{\text{pred}} \neq a$ **then**

11:    $\mathcal{E}_{\text{diverse}} \leftarrow \mathcal{E}_{\text{diverse}} \cup \{(q, a)\}$

12:    count $\leftarrow$ count $+ 1$

13:   **end if**

14:   **if** count $= k$ **then**

15:    **break**

16:   **end if**

17:  **end for**

18:  **if** count $< k$ **then**            ▷ Sample Random Correct Examples

19:   $\mathcal{E}_{\text{diverse}} \leftarrow \mathcal{E}_{\text{diverse}} \cup$ random.sample($S, k -$ count)

20:  **end if**

21:  **return** $\mathcal{E}_{\text{diverse}}$

22: **end procedure**

---

---

**Algorithm 5** SequentialOptimization Procedure

---

1: **Input:** $L$: large language model; $D$: problem description; $\hat{P}$: optimized prompt; $\mathcal{E}_{\text{diverse}} = \{(q_{d_i}, a_{d_i})\}_{i=1}^{k}$: diverse examples; $n$: *max_seq_iter*
2: **Output:** Final optimized task instruction $\hat{P}_{\text{opt}}$ and synthetic few-shot examples $\mathcal{E}_{\text{syn}} = \{(q_{s_i}, a_{s_i})\}_{i=1}^{k}$
3: **procedure** SEQUENTIALOPTIMIZATION($L, \hat{P}, \mathcal{E}_{\text{diverse}}, n$)
4:    $\mathcal{E}_{\text{syn}} \leftarrow \mathcal{E}_{\text{diverse}}$
5:    **for** round = 1 to $n$ **do**
6:       *feedback* $\leftarrow$ CritiqueComponent ($\hat{P}, \mathcal{E}_{\text{syn}}$)          ▷ Examples optimization step
7:       $\mathcal{E}_{\text{syn}} = \{(q_{s_i}, a_{s_i})\}_{i=1}^{k} \leftarrow$ SynthesizeComponent ($\mathcal{E}_{\text{diverse}},$ *feedback*)
8:
9:       *feedback* $\leftarrow$ CritiqueComponent ($\hat{P}, \mathcal{E}_{\text{syn}}$)          ▷ Prompt optimization step
10:      $\hat{P} \leftarrow$ SynthesizeComponent ($\hat{P}, \mathcal{E}_{\text{syn}},$ *feedback*)
11:   **end for**
12:   **return** $\hat{P}_{\text{opt}} \leftarrow \hat{P}, \mathcal{E}_{\text{syn}}$
13: **end procedure**

---

## 15 BEST PROMPTS FOR BBII TASKS

Below are the best prompt obtained using `PromptWizard` for some of the tasks in BBII dataset.

**antonyms**   Your task is to provide an antonym for each word presented to you, keeping in mind that the opposite word can often be formed by using prefixes or suffixes. If it's not possible to do so without altering the root word, choose a standalone antonym that widely resonates the opposite meaning in common contexts. The aim here is not to rule out standard methods of forming antonyms or to seek context-free opposites, but rather to find straightforward, widely accepted opposites based on every day usage and understanding. Regarding adverbs, note that some can have more than one antithesis depending on context, so provide the most generally applicable one. Ensure that the antonyms offered reflect commonly understood oppositions, without venturing into less accepted or contextually delicate nuances. Remember, the focus here is on providing clear, generally suitable opposites rather than unusual or highly situational counterparts.

For each input word present the reasoning followed by the correct word. Wrap only your final answer, without reason for each question separately between <ANS_START> and <ANS_END>.

**negation**   Initiate text inversion by transforming the sentiment of the input sentence to its exact reverse, while maintaining syntactic and grammatical accuracy and ensuring the output clearly communicates the opposing sentiment. Stick to input sentences that express opinions, feelings, or subjective judgments instead of factual, real-world information or historical events.

If the sentence contains an auxiliary verb, add the negation 'not' immediately after it. For sentences without an auxiliary, add 'not' before the main verb. If the input sentence includes a negative term, eliminate it to achieve the reverse sentiment.

Examine any clauses with modal verbs closely, keeping in mind to switch 'can' to 'can't' and so forth to reverse meaning. Be cautious while altering relative clauses, indirect speech, or idiomatic expressions. Their sentiment inversion should be handled carefully while still preserving linguistic coherence.

Consider implicit sentiments such as rhetorical questions, forms of irony, or sarcasm. Remember, altering these doesn't merely mean skewing negative to positive or vice versa. The key is to ensure clarity and comprehension of the reversed sentiment.

Avoid changing the truth value of objective facts or historical events, and if the main verb of a sentence doesn't carry the sentiment, consider implementing changes to other parts of the sentence—like the subject or object—to successfully reverse the meaning. Regularly assess the result of your modifications for precision and understanding."

For each input sentence, negate the meaning by adding 'not' to the input sentence. Wrap only your final answer, without reason for each question separately between <ANS_START> and <ANS_END>.

**second word letter**   For the provided word, your task is to specifically output the second letter.

For each input word, output only the extracted letter (only single letter) wrapped between <ANS_START> and <ANS_END> tags.

**sentence similarity**   For each input, you will find two sentences (Sentence 1 and Sentence 2). Your task is to evaluate their similarity based on two elements: overall meaning and specific numerical or factual details.

The importance of each element is weighted as follows: 70% overall meaning and 30% numerical/factual details.

The evaluation scale is now:

0 - Definitely not: The sentences not only differ in overall meaning but also show significant discrepancies in factual details. 1 - Probably not: There are minor similarities in meaning, but significant differences in factual details are prevalent. 2 - Possibly: The sentences share some elements of meaning but show differences in certain details or numerical data. 3 - Probably: The sentences express largely similar meanings but have noticeable differences or discrepancies in specific details or numerical data. 4 - Almost perfectly: The sentences are very similar in meaning with only slight discrepancies in factual or numerical details. 5 - Perfectly: The sentences are identical in terms of overall meaning and factual/numerical details.

In case of conflicts between overall meaning and factual details, the weighting system will guide your evaluation. Resultant rating should be separated with " - " for clarity, and should be accompanied by a brief textual description of your rating.

Provide your rating and brief textual description for each pair of sentences from the 6 options. (0 - Definitely not, 1 - Probably not, 2 - Possibly, 3 - Probably, 4 - Almost perfectly, 5 - Perfectly) Wrap only your final answer, without reason for each question separately between <ANS_START> and <ANS_END> tags.

**synonyms**   Your assignment involves identifying a list of synonyms for a provided word. These synonym should not only share the same basic meaning with the given word, but should also be able to replace the original word in most of its use cases without resulting in loss of meaning or causing the sentence to sound strange. For example, "report" could be a synonym for "account" as both can be used in similar business and financial situations while preserving the essence of the original use. Pay attention to the part of speech; a suitable synonym for a noun should also be a noun. Beware of false friends that evoke similar themes but are not true synonyms; "rest" seems related to "pillow," but one is a tangible object and the other an action or state, making them non-interchangeable. Prioritize synonyms that maintain the semantic richness of the original term, employ them regularly in similar contexts, and ensure they have the same connotation. Simplify your task by rejecting words that have only a minor relationship or those that are broader in meaning.

For each input word, output a list of synonym words. Wrap only your final answer, without reason for each question separately between <ANS_START> and <ANS_END> tags.

**word sorting**   Given a series of words in the task, your assignment is to reorder them in alphabetical order, prioritizing by the first letter of every word. Think step-by-step and consider the most efficient way to sort the words. Wrap the list of sorted words between <ANS_START> and <ANS_END>.

## 16   PROMPT TEMPLATES

The prompt template for `MutateComponent` is: `<problem description> <thinking style pool> <#style_variation_number> < instruction>`, where `< instruction>` guides `MutateComponent` to generate new mutated prompts by combining the problem description with thinking styles.

The prompt template for `ScoringComponent` is: `<mutated/improved prompts> <mini batch examples> < instruction>`, where `< instruction>` guides `ScoringComponent` to evaluate all mutated prompts against the examples in the mini-batch.

The prompt template for `CritiqueComponent` to get critique over prompt instruction is: `<best mutated prompt> <selected mini batch examples> < instruction>`, where < instruction> guides `CritiqueComponent` to provide feedback on how to improve the prompt instruction based on the selected examples.

The prompt template for `SynthesizeComponent` to refine prompt instruction is: `<best mutated prompt> <critique feedback> < instruction>`, where < instruction> guides `SynthesizeComponent` to generate an improved prompt using the critique feedback.

The prompt template for `CritiqueComponent` to get critique over few-shot examples is: The prompt template for `CritiqueComponent` is structured as follows: `<negative examples> <improved prompt> < instruction>`. This guides the `CritiqueComponent` to provide detailed feedback for improving examples. For `SynthesizeComponent`, the prompt template is `<synthesized examples> <improved prompt> < instruction>`, aiding in the synthesis and refinement of new examples.

The prompt template for `CritiqueComponent` follows this structure: `<synthesized examples> <improved prompt> < instruction>`, guiding the `CritiqueComponent` to provide detailed feedback for prompt improvement. For `SynthesizeComponent`, the prompt template is `<synthesized examples> <improved prompt> < instruction>`, assisting in the synthesis and refinement of new optimized prompts for the synthetic examples. Figure 6 demonstrates the critique feedback on the prompt alongside the refined optimized prompt. Prompt Templates used by different components are shown in Fig. 8

## 17    BEST PROMPTS

Best prompt found for each dataset are shown below:

### 17.1    GSM8K PROMPT

```
<the optimized prompt instruction>

Analyze the given real-world mathematical problem step-by-step,
    identifying key information, relationships between different pieces
    of data, and the context. Understand the structure of the problem,
    whether it involves a sequence of events or a comparison between
    different quantities. Keep track of all variables and quantities
    mentioned in the problem. Use appropriate mathematical operations and
     formulas, including addition, subtraction, multiplication, division,
     and more complex operations if required. Understand and handle
    indirect relationships and different units of measurement. Apply
    specific rules or conditions given in the problem. Make assumptions
    when information is not explicitly provided. Consider the order of
    operations when performing calculations. Understand the structure and
     properties of the data in the problem. Finally, verify your answer
    against the original problem to ensure it is logical and accurate.
```

```
<synthesized examples + reasoning chain>

[Question] Tim rides his bike back and forth to work for each of his 5
    workdays.  His work is 20 miles away.  He also goes for a weekend
    bike ride of 200 miles.    If he can bike at 25 mph how much time
    does he spend biking a week?
[Answer] 1. Identify the key pieces of information: Tim bikes to work and
     back for 5 days, his work is 20 miles away, he goes for a 200-mile
    bike ride on the weekend, and his biking speed is 25 mph.
2. Understand that the problem involves a sequence of events: Tim's daily
     commute to work and back, and his weekend bike ride.
3. Calculate the total distance Tim bikes to work and back in a week: 20
    miles to work * 2 (for the return trip) = 40 miles per day. Multiply
    this by 5 days: 40 miles/day * 5 days = 200 miles.
4. Add the distance of Tim's weekend bike ride to the total distance he
    bikes to work: 200 miles (work) + 200 miles (weekend) = 400 miles.
```

**MutateAgent:**

You are given a task description and a prompt instruction and different styles known as meta prompts:
[Task Description]: **<problem description>**
[Meta Prompt]: **<thinking style pool>**
Now you need to generate **<few_shot_count>** variations of following Instruction adaptively mixing meta prompt while keeping similar meaning.
Make sure to wrap each generated prompt with <START> and <END>
[Prompt Instruction]: **<agent instruction>**
[Generated Prompts]:

---

**SynthesizeAgent for refining instruction:**

I'm trying to write a zero-shot instruction that will help the most capable and suitable agent to solve the task.
My current prompt is: "**<agent instruction>**"
But this prompt gets the following examples wrong: **<negative examples>**
On carefully analysing these examples, following are the critiques related to prompt **<critic feedback>**
Use the critique smartly, refine the current prompt to make sure we don't get these examples wrong.
Based on the above information, Now I want you to write a different improved prompts.
Each prompt should be wrapped with <START> and <END>.
[Refined Prompts]:

---

**CriticAgent for few shot examples:**

You are an expert example selector who can help in selection of right in-context examples to help the most suitable agent solve this problem. You are also given the prompt instruction which is used to solve this task

[Prompt]: **<agent instruction>**
You are given the task description of the task:
[Task Description]: **<problem description>**
I'm trying to write a few shots prompt using **<few_shot_count>** in-context examples to effectively solve any questions of the above task.
My current **<few_shot_count>** in-context examples set are: {examples}
Think of analysing, understanding and creating examples of task on the criteria of diversity of types of examples, complexity of the nature/characteristics of the examples and relevance/compatibility to the whole example set in total.
Output all the suggestions/ improvement which could be made to improve each individual example of the whole example selection set.

---

**CriticAgent for instruction:**

I'm trying to write a zero-shot instruction that will help the most capable and suitable agent to solve the task.
My current prompt is: "**<agent instruction>**"
But this prompt gets the following examples wrong: **<negative examples>**
Provide detail feedback which identifies reasons where the instruction could have gone wrong.
Wrap each reason with <START> and <END>

---

**Generate keywords that express human intent:**

You are given an instruction along description of task labelled as [Task Description]. For the given instruction, list out 3-5 keywords in comma separated format as [Intent] which define the characteristics or properties required by the about the most capable and suitable agent to solve the task using the instruction.

[Task Description]: **<problem description>**
[Instruction]: **<agent instruction>**

[Intent]:

---

**SynthesizeAgent for generating examples:**

You are an expert example selector who can help in selection of right in-context examples to help the agent solve this problem.
You are also given the prompt instruction which is used to solve this task

[Prompt]: **<improved prompt>**
You are given the description of the task:
[Task Description]: **<problem description>**
I'm trying to write a few shots prompt using **<few_shot_count>** in-context examples to effectively solve any questions of the   above task
My current **<few_shot_count>** in-context examples set are: **<synthesized examples>**
You are also given a set of suggestions/improvements which could be made to improve each individual example of the whole example selection set:
[SUGGESTION/IMPROVEMENT]: **<critic feedback>**
Based on the above information, use all of it smartly and diligently to carefully create new set of **<few_shot_count>**, which follow these suggestion and improvements.
Make sure to output each example wrapped with <START> and <END>.

New examples should follow this format strictly:

[Question] followed by question part of the example
[Answer] followed by the all the steps of logic reasoning statements related to answer. The final answer as "<ANS_START>[answer]<ANS_END>"

For Example: <START>
**<original example>**
<END>

[New Examples]:

---

**Generate reasoning behind the answer:**

You are given a task description and instruction followed by a set of correct examples of the task.
[Task Description]: **<problem description>**
[Instruction]: **<agent instruction>**

Each example has a question denoted by question [Question] and a final answer [Answer] .
[Question]: **<question>**
[Answer]: **<answer>**

Now your task is to generate a reasoning chain that contains the steps, logical pathway followed to arrive at the correct answer,   assuming the necessary domain knowledge is present as part of the question and task description.
Make sure it is specific, non-ambiguous, complete, and specifies all the logic and steps required to reach the final answer.

[Improved Reasoning Chain]:

---

**Generate expert persona:**

For each instruction, write a high-quality description about the most capable and suitable agent to answer the instruction. In second person perspective.

[Instruction]:  Make a list of 5 possible effects of deforestation
[Agent Description]: You are an environmental scientist with a specialization in the study of ecosystems and their interactions with human activities. You have extensive knowledge about the effects of deforestation on the environment, including the impact on biodiversity, climate change, soil quality, water resources, and human health. Your work has been widely recognized and has contributed to the development of policies and regulations aimed at promoting sustainable forest management practices. You are equipped with the latest research findings, and you can provide a detailed and comprehensive list of the possible effects of deforestation, including but not limited to the loss of habitat for countless species, increased greenhouse gas emissions, reduced water quality and quantity, soil erosion, and the emergence of diseases. Your expertise and insights are highly valuable in understanding the complex interactions between human actions and the environment.
...

[Instruction]: **<problem description>**
[Agent Description]:

Figure 8: Prompt Templates for different components of `PromptWizard`.

5. Understand that the problem asks for the total time Tim spends biking in a week, and that time can be calculated by dividing distance by speed.

6. Calculate the total time Tim spends biking in a week: 400 miles / 25 mph = 16 hours.

7. Verify that the answer is logical: Tim spends 16 hours biking in a week, which is reasonable given the distances and speed provided.

8. The final answer is 16 hours. <ANS_START>16<ANS_END>

[Question] Tobias is buying a new pair of shoes that costs $95. He has been saving up his money each month for the past three months. He gets a $5 allowance a month. He also mows lawns and shovels driveways. He charges $15 to mow a lawn and $7 to shovel. After buying the shoes, he has $15 in change. If he mows 4 lawns, how many driveways did he shovel?

[Answer] 1. Identify the total amount of money Tobias had before buying the shoes. This is given by the cost of the shoes plus the change he has left, which is $95 + $15 = $110.

2. Calculate the total amount of money Tobias earned from his allowance. He gets $5 a month and has been saving for three months, so he earned $5 * 3 = $15 from his allowance.

3. Calculate the total amount of money Tobias earned from mowing lawns. He charges $15 to mow a lawn and he mowed 4 lawns, so he earned $15 * 4 = $60 from mowing lawns.

4. Subtract the money Tobias earned from his allowance and mowing lawns from the total amount of money he had before buying the shoes. This will give us the amount of money he earned from shoveling driveways. So, $110 − $15 − $60 = $35 is the amount he earned from shoveling driveways.

5. Finally, divide the total amount of money Tobias earned from shoveling driveways by the amount he charges to shovel one driveway. This will give us the number of driveways he shoveled. So, $35 / $7 = 5 driveways. <ANS_START>5<ANS_END>

[Question] Bella bought stamps at the post office. Some of the stamps had a snowflake design, some had a truck design, and some had a rose design. Bella bought 11 snowflake stamps. She bought 9 more truck stamps than snowflake stamps, and 13 fewer rose stamps than truck stamps. How many stamps did Bella buy in all?

[Answer] 1. Identify the quantities given in the problem: Bella bought 11 snowflake stamps.

2. Understand the relationships between the different types of stamps: She bought 9 more truck stamps than snowflake stamps, and 13 fewer rose stamps than truck stamps.

3. Calculate the number of truck stamps: The number of truck stamps is 11 (snowflake stamps) + 9 = 20.

4. Calculate the number of rose stamps: The number of rose stamps is 20 (truck stamps) − 13 = 7.

5. Add up all the stamps: The total number of stamps Bella bought is 11 (snowflake stamps) + 20 (truck stamps) + 7 (rose stamps) = 38.

6. Verify the answer: Check that the total number of stamps (38) matches the sum of the individual quantities of each type of stamp (11 snowflake stamps, 20 truck stamps, 7 rose stamps). The answer is correct. <ANS_START>38<ANS_END>

[Question] Tina makes $18.00 an hour. If she works more than 8 hours per shift, she is eligible for overtime, which is paid by your hourly wage + 1/2 your hourly wage. If she works 10 hours every day for 5 days, how much money does she make?

[Answer] 1. Identify the key information: Tina's hourly wage is \$18.00, she works 10 hours a day for 5 days, and overtime is calculated as the hourly wage plus half the hourly wage for hours worked over 8 hours in a shift.

31 2. Calculate the regular pay: Tina works 10 hours a day, but only 8 hours are considered regular hours. So, for 5 days, she works 8 hours/day * 5 days = 40 hours.

32 3. Multiply the regular hours by the hourly wage to get the regular pay: 40 hours * $18.00/hour = $720.00.

33 4. Calculate the overtime hours: Tina works 10 hours a day, so she has 10 hours/day - 8 hours/day = 2 hours/day of overtime. Over 5 days, this is 2 hours/day * 5 days = 10 hours of overtime.

34 5. Calculate the overtime wage: The overtime wage is the hourly wage plus half the hourly wage, so $18.00/hour + 0.5 * $18.00/hour = $27.00/hour.

35 6. Multiply the overtime hours by the overtime wage to get the overtime pay: 10 hours * $27.00/hour = $270.00.

36 7. Add the regular pay and the overtime pay to get the total pay: $720.00 + $270.00 = $990.00.

37 8. Verify the answer: Tina makes $990.00 if she works 10 hours a day for 5 days, with overtime pay for hours worked over 8 hours in a shift. This is logical and matches the original problem. <ANS_START>990<ANS_END>

38

39 [Question] Samanthas last name has three fewer letters than Bobbies last name. If Bobbie took two letters off her last name, she would have a last name twice the length of Jamies. Jamies full name is Jamie Grey. How many letters are in Samanthas last name?

40 [Answer] 1. Start by identifying the key pieces of information from the problem: Samanthas last name has three fewer letters than Bobbies last name, and if Bobbie took two letters off her last name, she would have a last name twice the length of Jamies. Jamies full name is Jamie Grey.

41 2. From the information given, we know that Jamie's last name is Grey, which has 4 letters.

42 3. Since Bobbie's last name, after removing two letters, is twice the length of Jamie's last name, we can set up the equation: (Bobbie's last name length - 2) = 2 * Jamie's last name length.

43 4. Substituting the known value of Jamie's last name length (4) into the equation gives: (Bobbie's last name length - 2) = 2 * 4, which simplifies to Bobbie's last name length - 2 = 8.

44 5. Solving for Bobbie's last name length gives: Bobbie's last name length = 8 + 2 = 10.

45 6. We know that Samanthas last name has three fewer letters than Bobbies last name. So, we can set up the equation: Samantha's last name length = Bobbie's last name length - 3.

46 7. Substituting the known value of Bobbie's last name length (10) into the equation gives: Samantha's last name length = 10 - 3.

47 8. Solving for Samantha's last name length gives: Samantha's last name length = 7.

48 9. Therefore, Samantha's last name has 7 letters. <ANS_START>7<ANS_END>

1 <human intent>

2

3 Mathematical Reasoning, Multi-step Problem Solving, Basic Arithmetic Operations, Data Analysis, Solution Verification

4

5 <expert persona>

6

7 You are a mathematics educator with a deep understanding of elementary and middle school mathematics. You are experienced in teaching multi-step problem-solving techniques and have a knack for breaking down complex problems into manageable steps. Your expertise lies in basic arithmetic operations such as addition, subtraction, multiplication, and division. You can provide clear, step-by-step solutions to mathematical problems that require multi-step reasoning. You are patient and thorough, ensuring that each step is clearly explained and understood. Your ability to simplify complex problems and guide

```
          students through the problem-solving process makes you an excellent
          resource for answering school-grade mathematical questions.
  8
  9 <prompt guidelines for desired output format>
  10
  11 You need to answer each of the questions separately. Arabic numeral
          should be at the end in the format Wrap your only the final answer,
          without reason for each question separately between <ANS_START> and <
          ANS_END> tags.
```

## 17.2  ETHOS PROMPT

```
  1 <the optimized prompt instruction>
  2
  3 Carefully examine the provided text and determine if it constitutes hate
          speech. Hate speech is defined as language that is derogatory,
          offensive, or discriminatory towards a specific group of people based
           on their race, gender, or other characteristics, with the intent to
          harm or degrade. Even if the statement appears factual or sarcastic,
          if it uses aggressive language or personal attacks aimed at a
          specific group, it should be classified as hate speech.
```

```
  1 <synthesized examples + reasoning chain>
  2
  3 [Question] While I appreciate the hard work of some immigrants, others
          are just here to exploit our system and take our jobs.
  4 [Answer] 1. Start by reading the provided text to understand its content.
  5 2. The task requires identifying if the text contains hate speech.
          According to the task description, hate speech is language that is
          derogatory, offensive, or discriminatory towards a specific group of
          people based on their race, gender, or other characteristics, with
          the intent to harm or degrade.
  6 3. The text states, "While I appreciate the hard work of some immigrants,
           others are just here to exploit our system and take our jobs."
  7 4. This statement is making a generalization about a specific group of
          people, in this case, immigrants. It suggests that some immigrants
          are exploiting the system and taking jobs, which is a negative
          stereotype.
  8 5. Even though the text acknowledges the hard work of some immigrants, it
           still uses derogatory language towards others, implying they are
          here to exploit the system.
  9 6. This statement can be seen as offensive and discriminatory towards
          immigrants, as it negatively stereotypes them and can potentially
          harm or degrade them.
  10 7. Therefore, according to the task description and instruction, this
          text can be classified as hate speech.\newline
  11 8. As per the task instruction, if the text contains hate speech, the
          output should be 1.
  12 9. Hence, the final answer is 1. <ANS_START>1<ANS_END>
  13
  14 [Question] I've noticed that female politicians are more corrupt and only
           care about their own interests.\newline
  15 [Answer] 1. Start by reading the provided text: "I've noticed that female
           politicians are more corrupt and only care about their own interests
          ."
  16 2. Identify the subject of the text: The subject is "female politicians."
  17 3. Analyze the language used in the text: The language used is derogatory
           and offensive, as it generalizes all female politicians as being "
          more corrupt" and only caring about their own interests.
  18 4. Determine if the language is discriminatory: The language
          discriminates against a specific group, in this case, female
          politicians, by making a negative generalization about them.
```

```
19  5. Check if the language has the intent to harm or degrade: The language
       used seems to degrade the reputation of female politicians by
       portraying them as corrupt and self-interested.
20  6. Based on the task description and instruction, the text is considered
       hate speech if it is derogatory, offensive, or discriminatory towards
        a specific group of people with the intent to harm or degrade.
21  7. Since the text meets all these criteria, it is classified as hate
       speech.
22  8. According to the task instruction, if the text contains hate speech,
       the output should be 1.
23  9. Therefore, the final answer is 1. <ANS_START>1<ANS_END>
24
25  [Question] I don't get why we have to accept people who choose to be
       transgender. It's unnatural and against our values.
26  [Answer] 1. Start by reading the provided text.
27  2. Identify the subject of the text. In this case, the subject is "people
        who choose to be transgender."
28  3. Analyze the language used in the text. The text uses phrases like "It'
       s unnatural" and "against our values," which are derogatory and
       offensive.
29  4. Determine if the language is aimed at a specific group. Here, the
       language is targeted towards transgender people, a specific group.
30  5. Check if the language is intended to harm or degrade the specific
       group. The text implies that being transgender is unnatural and
       against societal values, which can be seen as an attempt to degrade
       or harm the transgender community.\newline
31  6. Since the text uses offensive language against a specific group with
       the intent to harm or degrade, it can be classified as hate speech.
32  7. Therefore, the correct answer is 1, indicating that the text contains
       hate speech. <ANS_START>1<ANS_END>
```

```
1  <human intent>
2
3  Text Analysis, Hate Speech Identification, Discriminatory Language
      Detection, Offensive Content Recognition, Derogatory Language
      Classification
4
5  <expert persona>
6
7  You are a machine learning engineer with a specialization in natural
      language processing and text analysis. You have developed and trained
       algorithms to detect and classify different types of text, including
       hate speech. Your algorithm can analyze the given text, identify
      patterns, and determine whether it contains hate speech or not. You
      are equipped with a deep understanding of the nuances of language,
      and you can differentiate between offensive language, strong
      criticism, and actual hate speech. Your algorithm is designed to be
      fair and unbiased, and it respects the principles of freedom of
      speech while also recognizing the harm caused by hate speech. Your
      expertise in text analysis and machine learning is highly valuable in
       moderating online content and creating a safer digital environment.
8
9  <prompt guidelines for desired output format>
10 You need to answer each of the questions separately with 1 if its a hate
      speech and 0 if its not a hate speech.  Wrap your only the final
      answer, without reason for each question separately between <
      ANS_START> and <ANS_END> tags.
```

## 17.3 PUBMEDQA PROMPT

```
1  <the optimized prompt instruction>
2
3  Carefully read the abstract and identify the key points that directly
      address the biomedical research question. Based on these key points,
provide a clear and concise answer in a "Yes" or "No" format,
followed by a brief explanation that correlates the research question
 with the findings in the abstract.
```

```
 1  <synthesized examples + reasoning chain>
 2
 3  [Question] Does the process of photosynthesis under varying light
        intensities and temperature play a significant role in the growth of
        different species of algae?\newline
 4  [Abstract] : Photosynthesis is a process used by plants and other
        organisms to convert light energy into chemical energy that can later
         be released to fuel the organisms' activities. This study
        investigates the impact of varying light intensities and temperature
        on the photosynthetic process in different species of algae. The
        algae were exposed to different light intensities and temperatures,
        and their growth rate was monitored over a period of time. The
        results showed a direct correlation between light intensity,
        temperature and the growth rate of algae.
 5  [Answer] 1. The question asks whether the process of photosynthesis under
         varying light intensities and temperature plays a significant role
        in the growth of different species of algae.
 6  2. The abstract provides information about a study that investigates the
        impact of varying light intensities and temperature on the
        photosynthetic process in different species of algae.
 7  3. The abstract mentions that the algae were exposed to different light
        intensities and temperatures, and their growth rate was monitored
        over a period of time.\newline
 8  4. The results of the study, as mentioned in the abstract, showed a
        direct correlation between light intensity, temperature and the
        growth rate of algae.
 9  5. This direct correlation indicates that the process of photosynthesis
        under varying light intensities and temperature does indeed play a
        significant role in the growth of different species of algae.
10  6. Therefore, based on the information provided in the abstract, the
        answer to the question is "Yes". <ANS_START>yes<ANS_END>
11
12
13  [Question] Is the use of antiviral drugs effective in treating influenza,
         a common viral infection?
14  [Abstract] : Antiviral drugs are medicines used to prevent and treat
        viral infections. Influenza, on the other hand, is a viral infection.
         This study investigates the effectiveness of antiviral drugs in
        treating influenza. The study involved patients suffering from
        influenza who were treated with antiviral drugs. The results showed
        significant improvement in the condition of the patients.
15  [Answer] 1. The question asks about the effectiveness of antiviral drugs
        in treating influenza, a common viral infection.
16  2. The abstract provides information about a study that investigates the
        effectiveness of antiviral drugs in treating influenza.
17  3. The study involved patients suffering from influenza who were treated
        with antiviral drugs.\newline
18  4. The results of the study showed significant improvement in the
        condition of the patients after they were treated with antiviral
        drugs.
19  5. Therefore, based on the results of the study mentioned in the abstract
        , it can be concluded that the use of antiviral drugs is effective in
         treating influenza.
20  6. Hence, the answer to the question is "Yes". <ANS_START>yes<ANS_END>
21
22
23  [Question] Are intensive care units more beneficial than general wards
        for the treatment of severe pneumonia in children with underlying
        health conditions?
24  [Abstract] : Pneumonia is a common illness in children that can become
        severe if not properly treated. Intensive care units (ICUs) provide
```

```
      specialized care for patients with severe or life-threatening
      illnesses. This study examines the impact of ICU treatment on
      children with severe pneumonia and underlying health conditions. The
      study compared the recovery rates of children treated in ICUs with
      those treated in general wards. The results showed a higher recovery
      rate in children with underlying health conditions treated in ICUs.\
      newline
25 [Answer] 1. The question asks whether intensive care units (ICUs) are
      more beneficial than general wards for the treatment of severe
      pneumonia in children with underlying health conditions.
26 2. The abstract provides information about a study that examined the
      impact of ICU treatment on children with severe pneumonia and
      underlying health conditions.\newline
27 3. The study compared the recovery rates of children treated in ICUs with
       those treated in general wards.
28 4. The results of the study showed a higher recovery rate in children
      with underlying health conditions treated in ICUs.
29 5. Therefore, based on the results of the study presented in the abstract
      , the answer to the question is "Yes". ICUs are more beneficial than
      general wards for the treatment of severe pneumonia in children with
      underlying health conditions. <ANS_START>yes<ANS_END>
30
31 [Question] Is the blood glucose level a more reliable marker than HbA1c
      for diagnosing Diabetes?
32 [Abstract] : Diabetes is a chronic disease that affects the body's
      ability to process sugar. Blood glucose levels and HbA1c are commonly
       used markers for diagnosing diabetes. This study investigates the
      reliability of blood glucose levels and HbA1c as markers for
      diagnosing Diabetes. The study involved patients diagnosed with
      Diabetes and their blood glucose and HbA1c levels were measured. The
      results showed a significant correlation between high blood glucose
      levels and Diabetes diagnosis, but not with HbA1c levels.
33 [Answer] 1. The question asks whether blood glucose level is a more
      reliable marker than HbA1c for diagnosing Diabetes.\newline
34 2. The abstract provides information about a study that investigates the
      reliability of blood glucose levels and HbA1c as markers for
      diagnosing Diabetes.
35 3. The abstract mentions that the study involved patients diagnosed with
      Diabetes and their blood glucose and HbA1c levels were measured.
36 4. The key point in the abstract is the results of the study, which
      showed a significant correlation between high blood glucose levels
      and Diabetes diagnosis, but not with HbA1c levels.
37 5. This indicates that blood glucose levels are a more reliable marker
      for diagnosing Diabetes than HbA1c levels, according to the study.
38 6. Therefore, based on the information provided in the abstract, the
      answer to the question is "Yes". <ANS_START>yes<ANS_END>
39
40
41 [Question] Can regular strength training reduce the risk of osteoporosis
      in adults over 60?
42 [Abstract] : Osteoporosis is a major health issue globally, especially in
       adults over 60. Regular strength training is known to have various
      health benefits, including improving bone health. This study
      investigates the impact of regular strength training on the risk of
      osteoporosis in adults over 60. The study involved participants who
      engaged in regular strength training and their bone health was
      monitored over a period of time. The results showed a lower incidence
       of osteoporosis in participants who engaged in regular strength
      training.
43 [Answer] 1. The question asks whether regular strength training can
      reduce the risk of osteoporosis in adults over 60.
44 2. The abstract provides information about a study that investigates the
      impact of regular strength training on the risk of osteoporosis in
      adults over 60.
```

45 3. The abstract mentions that regular strength training is known to have
       various health benefits, including improving bone health.
46 4. The study involved participants who engaged in regular strength
       training and their bone health was monitored over a period of time.
47 5. The results of the study, as mentioned in the abstract, showed a lower
        incidence of osteoporosis in participants who engaged in regular
       strength training.
48 6. Therefore, based on the results of the study mentioned in the abstract
       , it can be concluded that regular strength training can reduce the
       risk of osteoporosis in adults over 60.
49 7. Hence, the answer to the question is "Yes". <ANS_START>yes<ANS_END>

1 <human intent>
2 Biomedical Research Understanding, Abstract Analysis, Key Point
       Identification, Concise Answering, Explanation Correlation
3
4 <expert persona>
5
6 You are a biomedical researcher with a deep understanding of medical and
       scientific literature. You have a strong background in reading and
       interpreting scientific abstracts, and you are skilled at extracting
       key information from complex texts. You can accurately answer
       biomedical research questions based on the information provided in
       the corresponding abstracts. Your expertise in biomedical research
       allows you to understand the nuances and implications of the findings
        presented in the abstracts, and you can provide clear, concise, and
       accurate answers to the questions. Your ability to critically analyze
        and interpret scientific literature makes you an invaluable resource
        in the field of biomedical research.
7
8 <prompt guidelines for desired output format>
9
10 You need to answer each of the questions separately with yes/ no/ maybe.
       Wrap your only the final answer, without reason for each question
       separately between <ANS_START> and <ANS_END> tags.

## 17.4 MedQA Prompt

1 <the optimized prompt instruction>
2
3 Analyze the patient's age, symptoms, duration and onset of symptoms,
       history of present illness, lifestyle factors, physical examination
       findings, and any diagnostic test results presented in the Medical
       Licensing Examination question. Use your knowledge of medicine to
       identify the most likely diagnosis or appropriate treatment. Consider
        the progression, severity, and duration of the patient's symptoms in
        relation to the answer options. Eliminate incorrect answer options
       based on your medical knowledge and ensure your final choice is the
       most appropriate given the specifics of the question. Validate your
       answer by ensuring it aligns with all the information provided in the
        question, including the patient's age, lifestyle factors, and
       specific diagnostic test results.

1 <synthesized examples + reasoning chain>
2
3 [Question] A 50-year-old man with a history of hypertension and type 2
       diabetes presents with a 3-day history of chest pain radiating to the
        left arm. He also reports shortness of breath and fatigue. Physical
       examination reveals a blood pressure of 150/90 mmHg, heart rate of
       90/min, and an irregular pulse. An ECG shows ST-segment elevation in
       leads II, III, and aVF. Which of the following is the most
       appropriate initial treatment?
4     Options:
5     A: Aspirin and clopidogrel

```
       6      B: Metformin
       7      C: Lisinopril
       8      D: Atorvastatin
       9
      10  [Answer] 1. Start by analyzing the patient's age, symptoms, and medical
              history. The patient is a 50-year-old man with a history of
              hypertension and type 2 diabetes. He presents with chest pain
              radiating to the left arm, shortness of breath, and fatigue. These
              symptoms are indicative of a cardiovascular event.
      11  2. Consider the physical examination findings. The patient has a blood
              pressure of 150/90 mmHg, heart rate of 90/min, and an irregular pulse
              . These findings further support the likelihood of a cardiovascular
              event.
      12  3. Review the diagnostic test results. The ECG shows ST-segment elevation
               in leads II, III, and aVF. This is a classic sign of an ST-segment
              elevation myocardial infarction (STEMI), a type of heart attack.
      13  4. Given the diagnosis of STEMI, consider the most appropriate initial
              treatment. The options are Aspirin and clopidogrel (A), Metformin (B)
              , Lisinopril (C), and Atorvastatin (D).
      14  5. Eliminate incorrect answer options based on medical knowledge.
              Metformin (B) is a medication for diabetes, Lisinopril (C) is an
              antihypertensive medication, and Atorvastatin (D) is a cholesterol-
              lowering medication. While these medications may be part of the
              patient's long-term management, they are not the most appropriate
              initial treatment for a STEMI.
      15  6. Aspirin and clopidogrel (A) are antiplatelet medications. They work by
               preventing blood clots, which is crucial in the initial management
              of a STEMI to restore blood flow to the heart muscle.
      16  7. Therefore, the most appropriate initial treatment for this patient,
              given his symptoms, physical examination findings, and ECG results,
              is Aspirin and clopidogrel (A). This aligns with all the information
              provided in the question and is the most appropriate given the
              specifics of the question.
      17  8. Validate the final choice (A) as it is the most appropriate initial
              treatment for a patient presenting with a STEMI. <ANS_START>A<ANS_END
              >
      18
      19
      20  [Question] A 6-month-old girl is brought to the physician by her mother
              because of a 2-day history of fever and irritability. She also has a
              rash on her cheeks. Physical examination reveals a temperature of
              38.5 C  (101.3 F ), a heart rate of 120/min, and a respiratory rate
              of 30/min. Examination of the skin shows erythema of the cheeks with
              sparing of the nasal bridge and perioral area. Which of the following
               is the most likely diagnosis?
      21      Options:
      22      A: Measles
      23      B: Fifth disease
      24      C: Roseola
      25      D: Scarlet fever
      26
      27  [Answer] 1. Start by analyzing the patient's age, symptoms, duration and
              onset of symptoms, and physical examination findings. The patient is
              a 6-month-old girl with a 2-day history of fever and irritability.
              She also has a rash on her cheeks. Her temperature is 38.5 C  (101.3
               F ), a heart rate of 120/min, and a respiratory rate of 30/min. The
              skin examination shows erythema of the cheeks with sparing of the
              nasal bridge and perioral area.
      28  2. Use your medical knowledge to identify the most likely diagnosis. The
              symptoms presented are indicative of a viral exanthem, a rash that
              appears due to a viral infection.
      29  3. Consider the answer options. The options are Measles, Fifth disease,
              Roseola, and Scarlet fever. All of these are diseases that can
              present with a rash.
```

4. Eliminate incorrect answer options based on your medical knowledge. Measles typically presents with a rash that starts at the hairline and moves down, along with Koplik spots in the mouth, which are not mentioned in the question. Scarlet fever typically presents with a sandpaper-like rash and a strawberry tongue, which are also not mentioned. Roseola typically presents with a high fever that suddenly drops as a rash appears, which does not match the patient's symptoms.

5. The remaining option is Fifth disease, also known as erythema infectiosum. This disease is common in children and presents with a "slapped cheek" rash, fever, and irritability, which aligns with the patient's symptoms.

6. Validate your answer by ensuring it aligns with all the information provided in the question. The patient's age, symptoms, and physical examination findings all align with a diagnosis of Fifth disease.

7. Therefore, the correct answer is B: Fifth disease. <ANS_START>B<ANS_END>

[Question] A 70-year-old man presents with a 1-year history of progressive memory loss, difficulty finding words, and getting lost in familiar places. Neurologic examination shows impaired recall and disorientation to time and place. MRI of the brain shows cortical atrophy and enlarged ventricles. Which of the following is the most likely diagnosis?
Options:
A: Alzheimer's disease
B: Vascular dementia
C: Lewy body dementia
D: Frontotemporal dementia

[Answer] 1. Start by analyzing the patient's age, symptoms, duration and onset of symptoms, and the results of the physical examination and diagnostic tests. The patient is a 70-year-old man with a 1-year history of progressive memory loss, difficulty finding words, and getting lost in familiar places. The neurologic examination shows impaired recall and disorientation to time and place. The MRI of the brain shows cortical atrophy and enlarged ventricles.

2. Consider the progression, severity, and duration of the patient's symptoms. The symptoms have been progressing over a year, which indicates a chronic condition.

3. Use your medical knowledge to identify the most likely diagnosis. The symptoms of progressive memory loss, difficulty finding words, and getting lost in familiar places, along with impaired recall and disorientation to time and place, are characteristic of a neurodegenerative disease.

4. Look at the answer options and eliminate incorrect ones based on your medical knowledge. Vascular dementia (Option B) typically presents with stepwise deterioration of cognitive function, which is not the case here. Lewy body dementia (Option C) is usually accompanied by visual hallucinations, parkinsonism, or fluctuating cognition, none of which are mentioned in the question. Frontotemporal dementia (Option D) often presents with changes in personality and behavior, which is also not mentioned in the question.

5. The remaining option is Alzheimer's disease (Option A), which is a neurodegenerative disease that commonly presents with progressive memory loss, difficulty finding words, and getting lost in familiar places, especially in older adults. The MRI findings of cortical atrophy and enlarged ventricles are also consistent with Alzheimer's disease.

6. Validate your answer by ensuring it aligns with all the information provided in the question. Alzheimer's disease fits with the patient's age, the chronic and progressive nature of the symptoms, the neurologic examination findings, and the MRI results.

7. Therefore, the correct answer is A: Alzheimer's disease. <ANS_START>A<ANS_END>

[Question] A 35-year-old woman presents with a 2-week history of severe headache, fever, and photophobia. She also reports a rash on her lower extremities. Physical examination reveals a temperature of 38.2 C (100.8 F ), a heart rate of 110/min, and a petechial rash on her lower extremities. Lumbar puncture shows increased white blood cells with a predominance of lymphocytes, increased protein, and normal glucose. Which of the following is the most appropriate pharmacotherapy?
Options:
A: Ceftriaxone and vancomycin
B: Acyclovir
C: Amphotericin B
D: Doxycycline

[Answer] 1. Start by analyzing the patient's symptoms: severe headache, fever, photophobia, and a petechial rash on her lower extremities. These symptoms suggest a systemic infection, possibly involving the central nervous system given the presence of headache and photophobia.
2. Consider the patient's age and duration of symptoms. A 35-year-old woman with a 2-week history of these symptoms suggests an acute infection rather than a chronic condition.
3. Review the physical examination findings and diagnostic test results. The patient has a fever and tachycardia, further supporting the presence of a systemic infection. The lumbar puncture results show increased white blood cells with a predominance of lymphocytes, increased protein, and normal glucose. These findings are indicative of viral meningitis.
4. Evaluate the answer options in relation to the most likely diagnosis. Viral meningitis is typically caused by enteroviruses, herpes simplex virus, or arboviruses.
5. Option A (Ceftriaxone and vancomycin) is used to treat bacterial meningitis, which is not consistent with the lumbar puncture results. Eliminate this option.
6. Option B (Acyclovir) is an antiviral medication used to treat infections caused by herpes viruses, including herpes simplex virus meningitis. This option aligns with the diagnosis.
7. Option C (Amphotericin B) is an antifungal medication, which is not consistent with the diagnosis of viral meningitis. Eliminate this option.
8. Option D (Doxycycline) is an antibiotic used to treat bacterial infections, including certain types of bacterial meningitis, but it is not the first-line treatment for viral meningitis. Eliminate this option.
9. Validate the final choice (Option B: Acyclovir) by ensuring it aligns with all the information provided in the question, including the patient's age, symptoms, physical examination findings, and specific diagnostic test results.
10. Therefore, the correct answer is B: Acyclovir. <ANS_START>B<ANS_END>

[Question] A 40-year-old man with a history of alcohol abuse presents with a 1-day history of severe abdominal pain, nausea, and vomiting. Physical examination reveals a distended abdomen, decreased bowel sounds, and tenderness to palpation in the upper abdomen. Laboratory tests show an elevated serum amylase and lipase. Which of the following is the most likely diagnosis?
Options:
A: Acute pancreatitis
B: Peptic ulcer disease
C: Gastric cancer

```
    D: Gastroenteritis

[Answer] 1. Start by analyzing the patient's age, symptoms, duration and
    onset of symptoms, history of present illness, lifestyle factors,
    physical examination findings, and any diagnostic test results
    presented in the question. The patient is a 40-year-old man with a
    history of alcohol abuse. He has been experiencing severe abdominal
    pain, nausea, and vomiting for 1 day. His abdomen is distended, bowel
    sounds are decreased, and there is tenderness in the upper abdomen.
    His serum amylase and lipase levels are elevated.
2. Use your knowledge of medicine to identify the most likely diagnosis.
    The patient's history of alcohol abuse, the sudden onset and severity
    of his symptoms, and his physical examination findings are all
    indicative of a pancreatic condition. The elevated serum amylase and
    lipase levels further support this, as these enzymes are produced by
    the pancreas and their levels increase in the blood when the pancreas
    is inflamed or damaged.
3. Consider the answer options in relation to the patient's symptoms and
    test results. Acute pancreatitis, peptic ulcer disease, gastric
    cancer, and gastroenteritis are all potential diagnoses.
4. Eliminate incorrect answer options based on your medical knowledge.
    Peptic ulcer disease typically presents with a burning pain in the
    middle or upper stomach between meals or at night, not with a
    distended abdomen and decreased bowel sounds. Gastric cancer usually
    develops slowly over many years, and its symptoms often only appear
    in the advanced stages of the disease. Gastroenteritis, while it can
    cause abdominal pain, nausea, and vomiting, does not typically result
    in a distended abdomen, decreased bowel sounds, or elevated serum
    amylase and lipase levels.
5. The remaining option, acute pancreatitis, aligns with all the
    information provided in the question. The patient's history of
    alcohol abuse is a common risk factor for acute pancreatitis. The
    sudden onset and severity of his symptoms, his physical examination
    findings, and his elevated serum amylase and lipase levels are all
    characteristic of this condition.
6. Therefore, the most likely diagnosis for this patient is acute
    pancreatitis, making option A the correct answer. <ANS_START>A<
    ANS_END>
```

```
<human intent>
Medical Knowledge, Analytical Skills, English Proficiency, Reasoning
    Skills, Attention to Detail

<expert persona>
You are a medical professional with extensive experience in the field and
     a deep understanding of the United States Medical Licensing Exam (
    USMLE). You have successfully passed the USMLE and have a thorough
    understanding of the format and style of the questions. You are well-
    versed in a wide range of medical topics, from anatomy and physiology
     to pathology and pharmacology. You have the ability to analyze
    complex medical scenarios, apply your knowledge, and make informed
    decisions. You can accurately interpret the questions and the
    provided options, and select the correct answer based on your medical
     knowledge and reasoning. Your expertise and experience make you
    highly capable of answering these questions correctly and efficiently

<prompt guidelines for desired output format>
You need to output the correct option among [A/B/C/D] for each question
    separately using your medical knowledge and reasoning. Wrap your only
     the final answer, without reason for each question separately
    between <ANS_START> and <ANS_END> tags.
```

