# OpenReview forum: "PromptWizard: Task-Aware Prompt Optimization Framework"
_ICLR.cc/2025/Conference — Submitted to ICLR 2025_

### Official Review · Reviewer_B1QR · 2024-10-23

**Soundness:** 3
**Presentation:** 2
**Contribution:** 2
**Rating:** 5
**Confidence:** 5

**Summary:**

This paper proposes a prompt optimization method based on Large Language Models (LLMs) named PromptWizard. PromptWizard not only optimizes instructions but also improves demonstrations by incorporating elements such as few-shot learning, Chain-of-Thought (CoT), task intent, and expert personas. In extensive experiments, PromptWizard achieves the best performance with minimal resource consumption.

**Strengths:**

1. The writing of the paper is clear, and the content is easy to understand.
2. Compared to other baseline methods, PromptWizard has shown significant performance improvement.
3. This research conducted extensive experimental validation, providing thorough and reliable evidence.

**Weaknesses:**

1. In fact, feedback-based prompt optimization methods are already quite mature, including approaches like APO, PromptAgent, and PE2. PromptWizard is not the first feedback-driven optimization method, and the authors overlooked related works of this nature in their writing. Therefore, I recommend the authors add related work of this type.

2. Following Weakness 1, many studies have already demonstrated the strong performance of feedback-based optimization methods. I believe that a comparison with methods like APO and PromptAgent is essential. I strongly recommend that the authors supplement their discussion with comparisons to these feedback-based methods.

3. Building on Weakness 2, I noticed that the meta prompt in PromptWizard is significantly heavier compared to methods like APO. Based on my recent review of relevant works, StraGo indicated that APO has the lowest cost. Thus, PromptWizard should include a comparison with APO in terms of cost and performance.

4. I think PromptWizard seems to be a mix of different prompt design techniques. PromptWizard integrates feedback mechanisms similar to those in APO, keyword extraction techniques from AutoHint, and also combines strategies such as Chain-of-Thought (CoT), few-shot learning, and expert personas. I believe your integration doesn’t delve deeply enough into the reasons for combining these techniques. This makes the paper somewhat lacking in terms of innovation. I strongly recommend that the authors provide additional explanation for why these modules are used. If they are included in PromptWizard solely because CoT, few-shot, and expert personas can improve LLM performance, I believe that is insufficient for an academic paper.

5. Writing Weakness ：

    a) In the cost analysis section, the authors should only present the results and corresponding analysis. The detailed estimation process is more appropriate for the appendix.

    b) In Figure 6, there is an inconsistency between the Synthetic Example and Feedback. Based on my experience, when optimizing examples using LLMs, it's highly likely that scores or percentages would be involved. However, the Synthetic Example completely overlooks this aspect. I believe this example does not effectively illustrate the impact of the feedback mechanism on the synthetic example. I suggest that the authors choose a more representative example that can more intuitively demonstrate the role of feedback in the prompt optimization process.

    c) In line 482, "Ablation on Different Base LLMs." I don't think this qualifies as an ablation study; this section describes the impact of different models on PromptWizard. I suggest changing the title.

Reference:

1. Automatic Prompt Optimization with “Gradient Descent” and Beam Search (https://arxiv.org/pdf/2305.03495)
2. PromptAgent: Strategic Planning with Language Models Enables Expert-level Prompt Optimization (https://arxiv.org/pdf/2310.16427)
3. Prompt Engineering a Prompt Engineer (https://arxiv.org/pdf/2311.05661)
4. StraGo: Harnessing Strategic Guidance for Prompt Optimization (https://arxiv.org/pdf/2410.08601)

**Questions:**

1. How many candidate prompts does PromptWizard retain in each iteration?

2. What do "self-evolving" and "self-adaptive" refer to? How are they implemented, and why are they beneficial for balancing exploration and exploitation?

3. How is the "Deeper Exploitation of Task Nuances" mentioned in line 159 implemented?

4. Please see Weakness section.

---

> ### Author Response · Authors · 2024-11-23
>
> >In fact, feedback-based prompt optimization methods are already quite mature, including approaches like APO, PromptAgent, and PE2. PromptWizard is not the first feedback-driven optimization method, and the authors overlooked related works of this nature in their writing. Therefore, I recommend the authors add related work of this type.
>
> ### Comparison with Feedback-based prompt optimization approaches:
>
> Thank you for this thoughtful comment. We recognize the importance of explicitly addressing comparisons with similar feedback-based prompt optimization approaches such as APO, PromptAgent, and PROMST. Below is a detailed articulation of how PromptWizard (PW) stands apart from these methods, highlighting its unique contributions and advantages:
>
> 1.  **APO vs. PW**:
> APO employs a gradient-based feedback loop where an initial prompt is refined iteratively using bandit algorithms. While this approach introduces feedback for prompt refinement, it is inherently limited by a lack of **diverse exploration**. APO does not mutate the prompt through different reasoning styles or perspectives, which can restrict its ability to handle complex tasks, especially when the initial prompt is suboptimal. This is critical for challenging datasets like GSM8k or AQUARAT.
>
> a.  In contrast, **PW leverages a feedback-driven critique-and-synthesis mechanism** that incorporates **diverse thinking styles** for mutation, ensuring a wide exploration space before refining prompts. This not only accelerates convergence but also enhances robustness in scenarios where the initial prompt is weak.
>
> b.  Additionally, APO’s reliance on gradients and its iterative nature result in higher computational costs. Our experiments show that APO typically requires 2–3 hours for prompt optimization, with significantly higher API calls and token usage compared to PW, which achieves optimized prompts in just 20–30 minutes.
>
> 2.  **PromptAgent vs. PW**:
> PromptAgent introduces Monte Carlo Tree Search (MCTS) for exploring multiple prompt variations and uses an external LLM for feedback at every branch of the tree. While this enhances exploration, **feedback is not used to guide the exploration directly** but instead evaluates multiple independent variations. This results in inefficiency, as the system evaluates a large number of branches without iterative refinement driving convergence.
>
> a.  PW, on the other hand, uses feedback to **systematically guide exploration**, refining one best prompt at each iteration while integrating it with diverse examples. This approach ensures that feedback meaningfully shapes the optimization process, improving computational efficiency. PromptAgent often requires 3–4 hours for prompt generation, whereas PW completes the process in under 30 minutes, with superior performance demonstrated across datasets.
>
> 3.  **PW’s Key Differentiators**:
>
> a.  **Dual Optimization**: PW optimizes both the **prompt instruction** and **in-context examples** in tandem, addressing a broader range of challenges than methods like APO and PromptAgent, which focus solely on instruction refinement.
>
> b.  **Feedback-Guided Exploration**: Unlike APO and PromptAgent, PW tightly integrates feedback into the exploration process, enabling more efficient and directed iterations.
>
> c.  **Efficiency and Versatility**: PW significantly reduces computational overhead while maintaining superior performance across a variety of complex datasets, as shown in our evaluations on GSM8k, AQUARAT, and others.
>
> We will incorporate this detailed discussion and corresponding quantitative comparisons into the revised manuscript to clarify PW’s novelty and advantages over existing feedback-based prompt optimization approaches.

---

> > ### Author Response · Authors · 2024-11-23
> >
> > >Following Weakness 1, many studies have already demonstrated the strong performance of feedback-based optimization methods. I believe that a comparison with methods like APO and PromptAgent is essential. I strongly recommend that the authors supplement their discussion with comparisons to these feedback-based methods.
> >
> > Thank you for highlighting this concern. In response, we provide a detailed comparative analysis between PW and feedback-based approaches such as **APO** and **PromptAgent.**
> >
> > We evaluated **PW** against **APO** and **PromptAgent** across four datasets: **GSM8k, AQUARAT, SVAMP, and ETHOS** (see Table A below). The key findings are summarized below:
> >
> > 1.   **Accuracy**: PW consistently achieves the **highest accuracy** across all datasets, significantly outperforming APO and PromptAgent, especially on complex tasks such as **GSM8k**. For instance, PW demonstrates superior handling of reasoning-heavy datasets where feedback-guided iterative refinement is essential.
> >
> > 2.  **Efficiency**: PW demonstrates **significant efficiency** advantages over both APO and PromptAgent. By tightly integrating feedback into a **critique-and-synthesize loop**, PW minimizes redundant computations while systematically improving prompts.
> >
> > 3.  **Token Usage**: While APO produces simpler prompts with slightly lower token usage, this comes at the cost of significantly reduced accuracy. PW balances token efficiency with performance, ensuring minimal token consumption without sacrificing quality.
> >
> > We will incorporate this detailed analysis and include corresponding results (e.g., Table A) in the revised manuscript to make these points clearer.
> >
> > ### Table below shows the accuracy and cost analysis comparison between PW and other feedback-based methods.
> >
> > | Dataset | Accuracy(high) | Accuracy(high)|Accuracy(high) |Accuracy(high) | API Calls(low)| API Calls(low) |API Calls(low)  | API Calls(low)|  Tokens(low)| Tokens(low) | Tokens(low)|Tokens(low) |
> > | ------- | --------------- | ----------- | ------------ | ------------ | ------------ | ------------ | ------------ |------------ |------------ |------------ |------------ |-----
> > |  | DSPy    | PromptAgent     | APO         | PW           | DSPy | PromptAgent | APO | PW | DSPy | PromptAgent | APO | PW |
> > | GSM8k   | 78.2            | 68.84       | 25.67        | **90** | 915 | 2115 | 8490 | **147** | 262 | 500 | **109** | 237 |
> > | AQUARAT | 55.1            | 56.67       | 20.12        | **58.2** | 920 | 2200 | 8500 | **112** | 326 | 875 | **125** | 200|
> > | SVAMP   | 77              | 78.67       | 75.25        | **82.3** | 2300 | 2111 | 8000 | **178** | 189 | 680 | **85** | 127 |
> > | ETHOS   | 84.1            | 84.25       | 80.62        | **89.4** | 660 | 2217 | 8200 | **80** | 175 | 417 | **55** | 190 |
> >
> >
> > >Building on Weakness 2, I noticed that the meta prompt in PromptWizard is significantly heavier compared to methods like APO. Based on my recent review of relevant works, StraGo indicated that APO has the lowest cost. Thus, PromptWizard should include a comparison with APO in terms of cost and performance.
> >
> > Feedback-based methods like APO work well for simpler tasks, producing lightweight prompts with lower token usage but at the cost of reduced accuracy on complex datasets. Our evaluation shows that while APO achieves \~25% accuracy on GSM8k and \~20% on AQUARAT, PromptWizard (PW) achieves 90% and 58.2% respectively, demonstrating superior performance. Furthermore, PW achieves these results using significantly fewer API calls (\~100 versus APO's \~8000), making it more efficient in the optimization phase. While PW's final prompts (\~200 tokens) are slightly larger than APO's (\~100 tokens), this added complexity ensures maximal performance while balancing cost efficiency.

---

> > > ### Author Response · Authors · 2024-11-23
> > >
> > > >I think PromptWizard seems to be a mix of different prompt design techniques. PromptWizard integrates feedback mechanisms similar to those in APO, keyword extraction techniques from AutoHint, and also combines strategies such as Chain-of-Thought (CoT), few-shot learning, and expert personas. I believe your integration doesn’t delve deeply enough into the reasons for combining these techniques. This makes the paper somewhat lacking in terms of innovation. I strongly recommend that the authors provide additional explanation for why these modules are used
> > >
> > > ### Comparison with SOTA:
> > >
> > > PromptWizard (PW) introduces a unique framework that differentiates itself from evolutionary algorithms like PromptBreeder and EvoPrompt, as well as feedback-based systems such as APO and PromptAgent.
> > >
> > > **Comparison with Evolutionary Algorithms:**
> > > PromptBreeder and EvoPrompt rely on evolutionary strategies to generate and refine prompts through random mutation and recombination. While these methods explore diverse prompt variants, they lack a systematic mechanism to evaluate and refine prompts iteratively based on feedback or task-specific performance. In contrast, PW combines feedback-driven optimization with diverse exploration, ensuring that each iteration improves performance by systematically critiquing and refining prompts and examples. This leads to more efficient and targeted optimization compared to the more stochastic nature of evolutionary approaches.
> > >
> > > **Comparison with Feedback-Based Systems:**
> > > Systems like APO and PromptAgent utilize feedback for prompt refinement but are limited in scope. APO focuses on editing and regenerating prompts using feedback gradients, often struggling with complex tasks due to a lack of diverse exploration. PromptAgent employs Monte Carlo Tree Search for prompt refinement, which is computationally expensive and decouples feedback from exploration. PW addresses these limitations by tightly integrating feedback into its iterative refinement process, enabling more guided exploration of both prompts and examples, leading to significant gains in accuracy and efficiency.
> > >
> > > **Key Novelty and Innovation of PW:**
> > > While feedback and iterative prompt refinement are established concepts, PW extends their applicability by innovating in two key areas: (1) efficient and systematic prompt optimization tailored for task-specific performance and (2) joint optimization of prompts and examples, which is critical for tasks requiring nuanced context. These innovations make PW particularly effective for complex, multi-step reasoning tasks, as demonstrated by its superior performance across diverse datasets.
> > >
> > > ### Ablation Studies
> > >
> > > We have conducted detailed experiments isolating key building blocks of PromptWizard (PW) to evaluate their individual contributions. These ablations reinforce the significance of PW’s modular components—stage-wise optimization, synthesis of examples, and reasoning steps—highlighting their complementary roles in achieving superior task performance.
> > >
> > > The experiments, performed on the GSM8k dataset with a 2-shot setting, illustrate the impact of stage-wise optimization, synthesized examples, and reasoning steps. Results show that **Stage 1 (only prompt optimization)** achieves the lowest accuracy. Adding **Stage 2 (joint optimization of prompts and examples)** slightly improves results. The best performance is achieved by combining all components, including synthesized examples and reasoning steps, which highlight their complementary roles.
> > >
> > > | **Experiment**                         | **Description**                                              | **Accuracy** |
> > > |-----------------------------------------|--------------------------------------------------------------|--------------|
> > > | **Expt 1**                              | Stage 1: Only prompt optimization                            | 0.52         |
> > > | **Expt 2**                              | Stage 2: Joint optimization without examples (zero-shot)     | 0.65         |
> > > | **Expt 3**                              | Stage 1 + Stage 2 without examples (zero-shot)               | 0.73         |
> > > | **Expt 4**                              | Stage 2 with examples (no synthesis)                         | 0.70         |
> > > | **Expt 5**                              | Stage 1 + Stage 2 with examples (no synthesis)               | 0.77         |
> > > | **Expt 6**                              | Stage 2 with synthesized examples                            | 0.71         |
> > > | **Expt 7**                              | Stage 1 + Stage 2 with synthesized examples                  | 0.79         |
> > > | **Expt 8**                              | Stage 1 + Stage 2 + Examples (no synthesis) + CoT reasoning  | 0.80         |
> > > | **Expt 9**                              | Stage 1 + Stage 2 + Examples (with synthesis) + CoT reasoning| 0.83         |

---

> > > > ### Author Response · Authors · 2024-11-23
> > > >
> > > > >Writing: a) In the cost analysis section, the authors should only present the results and corresponding analysis. The detailed estimation process is more appropriate for the appendix.
> > > >
> > > > Thank you for your suggestion. We will revise the text to make it easier to read and follow.
> > > >
> > > > >Writing: b) In Figure 6, there is an inconsistency between the Synthetic Example and Feedback. Based on my experience, when optimizing examples using LLMs, it's highly likely that scores or percentages would be involved. However, the Synthetic Example completely overlooks this aspect. I believe this example does not effectively illustrate the impact of the feedback mechanism on the synthetic example. I suggest that the authors choose a more representative example that can more intuitively demonstrate the role of feedback in the prompt optimization process.
> > > >
> > > > In Figure 6, the original example included basic multiplication and addition. The feedback provided was to increase the complexity by incorporating fractions or percentages, and this feedback was utilized to create a new example requiring fractions to solve the problem instead of just basic arithmetic. There is no error or inconsistency in this example. However, we agree that it remains a simple illustration of synthesized examples. We will include more complex examples in the revised paper to better demonstrate the impact of the feedback mechanism. We hope this addresses your concern.
> > > >
> > > > >Writing c) In line 482, "Ablation on Different Base LLMs." I don't think this qualifies as an ablation study; this section describes the impact of different models on PromptWizard. I suggest changing the title.
> > > >
> > > > We will change the title and we have conducted additional experiments to show the impat of various stages of the framework. Please see earlier comment.
> > > >
> > > >
> > > > >Q1. How many candidate prompts does PromptWizard retain in each iteration?
> > > >
> > > > In each iteration, PromptWizard generates and mutates 9 candidate prompt instructions. These are scored to identify the best-performing prompt, which is then used to critique and synthesize a new prompt. This process continues until 3-5 iterations are completed, as defined in Table 11. By the end of each iteration, only the best prompt, incorporating feedback, is retained.
> > > >
> > > > >Q2. What do "self-evolving" and "self-adaptive" refer to? How are they implemented, and why are they beneficial for balancing exploration and exploitation?
> > > >
> > > > In **PromptWizard (PW)**:
> > > >
> > > > *   **Self-Evolving**: The LLM generates diverse new prompts by introducing variations using different reasoning styles, effectively "evolving" prompts to explore the search space comprehensively. This ensures a wide range of candidate prompts to test for effectiveness.
> > > > *   **Self-Adaptive**: The framework refines and aligns prompts based on task-specific feedback through a critique-and-synthesis process. This feedback-driven adaptation ensures optimized prompts tailored to improve performance.
> > > >
> > > > **Balancing Exploration and Exploitation**
> > > >
> > > > PW balances **exploration** by using self-evolution to diversify prompts and **exploitation** by self-adapting prompts that perform well. This dynamic ensures that the framework does not converge prematurely on suboptimal solutions while also maximizing the potential of high-performing prompts.
> > > >
> > > > This systematic balance drives PW’s robust performance in optimizing prompts and examples, leading to superior task outcomes.
> > > >
> > > > > Q3. How is the "Deeper Exploitation of Task Nuances" mentioned in line 159 implemented?
> > > >
> > > > Thank you for the thoughtful question. The "Deeper Exploitation of Task Nuances" is implemented during **Stage 2: Joint Optimization of Prompt and Examples** in the PromptWizard (PW) framework.
> > > >
> > > > 1.  **Feedback Integration from Examples**:
> > > >
> > > > *   In Stage 2, the refined prompt from Stage 1 is paired with selected examples, and both are optimized together. Feedback from the selected examples highlights areas where the prompt underperforms, providing specific task-related gradients for improvement.
> > > >
> > > > 2.  **Incorporating Task-Specific Nuances**:
> > > >
> > > > *   For instance, in GSM8k, the initial prompt failed to address nuances like handling relationships and units of measurement in math problems. Feedback from failed examples pinpointed these gaps, and the critique-and-synthesis process seamlessly incorporated these nuances into the prompt, significantly improving performance (see Section 17.1 lines 1114-1115).
> > > >
> > > > By optimizing prompts with feedback from diverse examples, PW systematically captures task-specific nuances, enhancing adaptability and robustness.
> > > >
> > > > **We have thoroughly addressed your concerns regarding novelty, comparisons with feedback-based systems, and additional ablation experiments. We believe these clarifications effectively resolve your concerns, and we look forward to receiving positive feedback on our work.**

---

> ### Comment · Reviewer_B1QR · 2024-11-25
> **Official Comment by Reviewer B1QR**
>
> Sorry for the late reply. Thank you very much for your explanation. However, I believe that PromptWizard still has some shortcomings in its methodological design, offering limited inspiration for subsequent research on prompt optimization. Nevertheless, I appreciate your efforts. I have adjusted my score and hope you can make corresponding revisions based on the valuable suggestions.

---

> > ### Author Response · Authors · 2024-11-25
> >
> > Thank you for your feedback and for revisiting your assessment of our work. We appreciate your suggestions and will incorporate the recommended changes into our revised manuscript.
> >
> > PromptWizard demonstrates that by applying the proposed techniques, it is possible to achieve superior performance across various tasks and domains while remaining computationally efficient. This is particularly critical in today's context, where writing effective prompts can take weeks or months of manual effort, especially as models evolve and task requirements shift. In contrast, PromptWizard completes this process in under 30 minutes, significantly reducing the time and effort required.
> >
> > Additionally, we plan to open-source the codebase, empowering the community to build upon and extend our framework for broader and evolving applications.

---

### Official Review · Reviewer_Vorh · 2024-11-01

**Soundness:** 2
**Presentation:** 2
**Contribution:** 2
**Rating:** 3
**Confidence:** 3

**Summary:**

The authors proposed an automated framework for discrete prompt optimization to address the challenges of labor-intensive and domain-specific manual prompt engineering. PromptWizard employs a self-adapting mechanism that uses a feedback-driven critique and synthesis process to iteratively refine prompt instructions and in-context examples. They demonstrated the effectiveness of PromptWizard across various tasks and highlighted the method's efficiency through a detailed cost analysis.

**Strengths:**

1. They performed extensive experiments on benchmark tasks like BBII, BBH, and arithmetic reasoning tasks.
2. The cost analysis demonstrates significant reductions in API calls, token usage, and overall costs, showcasing PromptWizard’s efficiency.

**Weaknesses:**

1. Readability is quite poor, and their contributions should be organized more clearly.  It looks like a list of various elements that were put into the algorithm for now.
2. It is difficult to identify the novelty of this work. It appears that they have integrated various existing methods. For instance, their iterative prompt refinement algorithm seems to be a combination of established techniques that originally introduced mutation, scoring, critique, and synthesis. Additionally, the self-adapting mechanism they mention is also widely used.

**Questions:**

1. Could you clarify the explanation of "identification of diverse examples"? What role do positive examples and negative examples play in this process?
2. Is there a specific reason why you only compared Instinct with PW in the one-shot setting, as opposed to the zero-shot setting experiment?

---

> ### Author Response · Authors · 2024-11-23
> **Novelty of the work**
>
> ## PromptWizard Contributions:
>
> PromptWizard (PW) achieves its superior performance through three core innovations that collectively differentiate it from existing approaches, ensuring both high accuracy and cost-efficiency.
>
> 1.   **Feedback-Driven Critique-and-Synthesis Mechanism**:
> PW introduces a novel feedback loop where an LLM generates, critiques, and refines prompts and examples iteratively. This mechanism balances **exploration** (through diverse mutations) and **exploitation** (guided feedback to converge on optimal prompts). Unlike evolutionary approaches such as PromptBreeder and EvoPrompt, which rely on mutation without structured feedback—leading to uncontrolled exploration and higher costs—PW integrates LLM-driven feedback at every stage to guide prompt refinement. This feedback loop ensures edits align with task requirements, reducing unnecessary evaluations and delivering better results with significantly lower computational costs. Compared to feedback-based systems like APO and PromptAgent, which lack diverse initial mutations, PW expands the search space by generating diverse prompts before applying feedback, preventing premature convergence to suboptimal solutions. This balanced and iterative process ensures task-relevant optimization while reducing computational overhead.
>
> 2.   **Sequential Optimization of Instructions and Examples**:
> PW uniquely optimizes instructions and examples in tandem through iterative refinement:
>
> a.   **Prompt Optimization with Examples**: The refined prompt undergoes evaluation with selected examples, and feedback informs further improvements.
>
> b. **Example Synthesis**: Using feedback, PW synthesizes diverse, task-relevant examples that enhance performance.
> By jointly optimizing prompts and examples, PW surpasses methods that treat these elements independently, creating cohesive, task-aware prompts.
>
> 3.  **Incorporation of CoT Reasoning**:
> PW integrates Chain-of-Thought (CoT) reasoning during optimization. This step-by-step reasoning framework leverages the refined prompts to guide the LLM in solving tasks effectively, further improving task performance and interpretability.
>
> **Takeaway**
>
> The key insight is that PW combines feedback-driven critique and synthesis, tandem optimization of prompts and examples, and self-generated reasoning steps into an automated, efficient framework. These elements are fundamental to PW's ability to outperform state-of-the-art methods while being significantly more cost-effective.
>
> While certain components may seem familiar, PW innovates by integrating these techniques into a cohesive system that automates and optimizes prompt engineering effectively. This work lays the foundation for future developments in prompt optimization by demonstrating that these techniques can be adapted and extended to address evolving challenges in this field.
>
> ## Comparison with SOTA:
>
> PromptWizard (PW) introduces a unique framework that differentiates itself from evolutionary algorithms like PromptBreeder and EvoPrompt, as well as feedback-based systems such as APO and PromptAgent.
>
> **Comparison with Evolutionary Algorithms:**
> PromptBreeder and EvoPrompt rely on evolutionary strategies to generate and refine prompts through random mutation and recombination. While these methods explore diverse prompt variants, they lack a systematic mechanism to evaluate and refine prompts iteratively based on feedback or task-specific performance. In contrast, PW combines feedback-driven optimization with diverse exploration, ensuring that each iteration improves performance by systematically critiquing and refining prompts and examples. This leads to more efficient and targeted optimization compared to the more stochastic nature of evolutionary approaches.
>
> **Comparison with Feedback-Based Systems:**
> Systems like APO and PromptAgent utilize feedback for prompt refinement but are limited in scope. APO focuses on editing and regenerating prompts using feedback gradients, often struggling with complex tasks due to a lack of diverse exploration. PromptAgent employs Monte Carlo Tree Search for prompt refinement, which is computationally expensive and decouples feedback from exploration. PW addresses these limitations by tightly integrating feedback into its iterative refinement process, enabling more guided exploration of both prompts and examples, leading to significant gains in accuracy and efficiency.
>
> **Key Novelty and Innovation of PW:**
> While feedback and iterative prompt refinement are established concepts, PW extends their applicability by innovating in two key areas: (1) efficient and systematic prompt optimization tailored for task-specific performance and (2) joint optimization of prompts and examples, which is critical for tasks requiring nuanced context. These innovations make PW particularly effective for complex, multi-step reasoning tasks, as demonstrated by its superior performance across diverse datasets.

---

> > ### Author Response · Authors · 2024-11-23
> >
> > >Q1. Could you clarify the explanation of "identification of diverse examples"? What role do positive examples and negative examples play in this process?
> >
> > Thank you for this question. We will add the below details in the revised paper for clarity. We hope this answers your question. PromptWizard (PW) employs a two-stage pipeline:
> >
> > 1.  **Stage 1: Iterative Prompt Optimization**
> >     In this stage, diverse initial prompts are generated and iteratively refined through a feedback-driven process, focusing solely on improving the instruction quality. This ensures the final prompt is well-optimized before proceeding to the next stage.
> >
> > 2.  **Stage 2: Joint Optimization of Prompt and Examples**
> >     In this stage, the refined prompt from Stage 1 is paired with selected examples, and both are optimized together.
> >
> > **Identification of Diverse Examples:**
> > In Stage 2 of PW, diverse examples are identified to ensure the prompt and examples are optimized comprehensively across various scenarios in the dataset. To achieve this, PW selects an initial set of examples using two methods:
> >
> > *   **Random Selection**: Ensures a broad, unbiased representation from the dataset.
> > *   **Clustering-Based Selection**: Groups data points with similar characteristics and selects examples from different clusters to cover diverse input scenarios.
> >
> > This process ensures that the selected examples effectively represent the dataset's variety, covering distinct cases that the model might encounter during inference.
> >
> > **Role of Positive and Negative Examples:**
> > Once examples are selected, PW evaluates their performance with the current prompt using a scoring mechanism. This evaluation uses metrics such as accuracy, F1 score, or LLM-based assessments for correctness and relevance. Based on their scores:
> >
> > *   **Positive Examples** (well-performing): These reinforce alignment between the prompt and task, serving as strong references to enhance the overall optimization.
> > *   **Negative Examples** (poorly performing): These highlight gaps in performance, enabling the critique-and-synthesis mechanism to refine prompts and generate new examples that address these shortcomings.
> >
> > The systematic integration of both positive and negative examples ensures that PW not only capitalizes on strengths but also actively resolves weaknesses, resulting in a robust and highly optimized framework.
> >
> > >Q2. Is there a specific reason why you only compared Instinct with PW in the one-shot setting, as opposed to the zero-shot setting experiment?
> >
> > We compared Instinct with PW in the one-shot setting because other algorithms, such as APE, InsZero, PromptBreeder, and EvoPrompt, are designed exclusively for zero-shot settings. These algorithms do not include mechanisms to optimize in-context examples within their prompts, making a direct comparison in a one-shot setting irrelevant.

---

> > > ### Author Response · Authors · 2024-11-25
> > >
> > > Thank you again for your feedback and thoughtful questions! As we are now approaching the end of the rebuttal period, I’m following up again to check if you have any further questions regarding our responses. We hope we have addressed your concerns and clarifications thoroughly and looking forward to a positive response and score.

---

> > > > ### Author Response · Authors · 2024-11-25
> > > >
> > > > Thank you again for your feedback and thoughtful questions! As we are now approaching the end of the rebuttal period, I’m following up again to check if you have any further questions regarding our responses. We hope we have addressed your concerns and clarifications thoroughly and looking forward to a positive response and score.

---

> > > > > ### Author Response · Authors · 2024-11-29
> > > > >
> > > > > Thank you again for your feedback and thoughtful questions! As we are now approaching the end of the rebuttal period, I’m following up again to check if you have any further questions regarding our responses. We hope we have addressed your concerns and clarifications thoroughly and looking forward to a positive response and score.

---

> > > > > > ### Author Response · Authors · 2024-12-02
> > > > > >
> > > > > > Thank you again for your feedback and thoughtful questions! As we are now approaching the end of the rebuttal period, I’m following up again to check if you have any further questions regarding our responses. We hope we have addressed your concerns and clarifications thoroughly and looking forward to a positive response and score.

---

### Official Review · Reviewer_hKKu · 2024-11-02

**Soundness:** 2
**Presentation:** 2
**Contribution:** 2
**Rating:** 3
**Confidence:** 4

**Summary:**

The paper proposes a prompt optimization framework to query LLM itself to automatically evolve, reflect, and optimize the prompt in the discrete space. One innovation part is that the framework optimizes both the head prompt and few-shot examples sequentially for holistic effects. The experiments over BBH and GSM8K shows better performance over other methods, and the much lower costs in API calls and token costs.

The paper has several problems: 1) The main framework is mostly close to other works such as APO, PROMST, and PromptAgent. However, the paper did not mention these works and compare with them. 2) Critique and feedback over the errors have been claimed as one innovation point in the paper. However, this component already appears in above three works. This paper may lack correct clarification on contributions. 3) The writing is not satisfying. Sometimes very hard to follow.

**Strengths:**

The optimization of head prompt and few-shot example prompt together is one novel component. I have not seen this setting in other papers before. The experiments over BBH and GSM8K and some ablation studies are appreciated.

**Weaknesses:**

1) Overclaim of the innovation part. Actually, integrating error feedback for LLMs to refine the prompts has already been proposed in other works, such as APO (https://openreview.net/pdf?id=WRYhaSrThy), PromptAgent (https://arxiv.org/pdf/2310.16427) and PROMST (https://arxiv.org/pdf/2402.08702), etc.
2) Following above, the authors only compare with baselines like Evoprompt, PromptBreeder, and APE, which truly do not use error feedback. However, why not compare with works using error feedback such as APO? I am curious that these frameworks can achieve similar performance and low costs.
3) The shared anonymized code link cannot be accessed.
4) The specific setting of baseline methods is not illustrated. How to make sure a fair comparison like the same number of prompt sampling?
5) Many components lack analysis of why it works and lack the study over the robustness of this framework. For example, whether the wording of component prompt will influence the performance.

**Questions:**

In the optimization of few-shot examples, how to make sure the examples do not contain errors if relying on LLMs to synthesize?

The framework relies heavily on LLM capability to refine and reflect. Is there any components that smaller LLMs cannot work well so that the framework fails?

---

> ### Author Response · Authors · 2024-11-22
>
> >> Overclaim of the innovation part. Actually, integrating error feedback for LLMs to refine the prompts has already been proposed in other works, such as APO ([https://openreview.net/pdf?id=WRYhaSrThy](https://openreview.net/pdf?id=WRYhaSrThy)), PromptAgent ([https://arxiv.org/pdf/2310.16427](https://arxiv.org/pdf/2310.16427)) and PROMST ([https://arxiv.org/pdf/2402.08702](https://arxiv.org/pdf/2402.08702)), etc.
> >> The main framework is mostly close to other works such as APO, PROMST, and PromptAgent. However, the paper did not mention these works and compare with them.
>
> Thank you for this comment. We will provide details of our contributions and also how PW compares to other feedback based approaches like APO and PromptAgent next. We hope this would clarify any concerns on the innovations and its performance compared to existing approaches.
>
> ### PromptWizard Contributions:
>
> PromptWizard (PW) achieves its superior performance through three core innovations that collectively differentiate it from existing approaches, ensuring both high accuracy and cost-efficiency. These insights address the limitations of prior work and offer a transferable foundation for future advancements:
>
> 1. **Feedback-Driven Critique-and-Synthesis Mechanism**:
> PW introduces a novel feedback loop where an LLM generates, critiques, and refines prompts and examples iteratively. This mechanism balances **exploration** (through diverse mutations) and **exploitation** (guided feedback to converge on optimal prompts). Unlike evolutionary approaches such as PromptBreeder and EvoPrompt, which rely on mutation without structured feedback—leading to uncontrolled exploration and higher costs—PW integrates LLM-driven feedback at every stage to guide prompt refinement. This feedback loop ensures edits align with task requirements, reducing unnecessary evaluations and delivering better results with significantly lower computational costs. Compared to feedback-based systems like APO and PromptAgent, which lack diverse initial mutations, PW expands the search space by generating diverse prompts before applying feedback, preventing premature convergence to suboptimal solutions. This balanced and iterative process ensures task-relevant optimization while reducing computational overhead.
>
> 2.  **Sequential Optimization of Instructions and Examples**:
> PW uniquely optimizes instructions and examples in tandem through iterative refinement:
>
> a.  **Prompt Optimization with Examples**: The refined prompt undergoes evaluation with selected examples, and feedback informs further improvements.
>
> b. **Example Synthesis**: Using feedback, PW synthesizes diverse, task-relevant examples that enhance performance.
> By jointly optimizing prompts and examples, PW surpasses methods that treat these elements independently, creating cohesive, task-aware prompts.
>
> 3. **Incorporation of CoT Reasoning**:
> PW integrates Chain-of-Thought (CoT) reasoning during optimization. This step-by-step reasoning framework leverages the refined prompts to guide the LLM in solving tasks effectively, further improving task performance and interpretability.
>
> **Takeaway**
>
> The key insight is that PW combines feedback-driven critique and synthesis, tandem optimization of prompts and examples, and self-generated reasoning steps into an automated, efficient framework. These elements are fundamental to PW's ability to outperform state-of-the-art methods while being significantly more cost-effective.
>
> While certain components may seem familiar, PW innovates by integrating these techniques into a cohesive system that automates and optimizes prompt engineering effectively. This work lays the foundation for future developments in prompt optimization by demonstrating that these techniques can be adapted and extended to address evolving challenges in this field.

---

> > ### Author Response · Authors · 2024-11-22
> > **Comparison with Feedback-based prompt optimization approaches:**
> >
> > Thank you for this thoughtful comment. We recognize the importance of explicitly addressing comparisons with similar feedback-based prompt optimization approaches such as APO, PromptAgent, and PROMST. Below is a detailed articulation of how PromptWizard (PW) stands apart from these methods, highlighting its unique contributions and advantages:
> >
> > 1. **APO vs. PW:**
> > APO employs a gradient-based feedback loop where an initial prompt is refined iteratively using bandit algorithms. While this approach introduces feedback for prompt refinement, it is inherently limited by a lack of diverse exploration. APO does not mutate the prompt through different reasoning styles or perspectives, which can restrict its ability to handle complex tasks, especially when the initial prompt is suboptimal. This is critical for challenging datasets like GSM8k or AQUARAT.
> >
> > **a.**	In contrast, **PW leverages a feedback-driven critique-and-synthesis mechanism that incorporates diverse thinking styles for mutation**, ensuring a wide exploration space before refining prompts. This not only accelerates convergence but also enhances robustness in scenarios where the initial prompt is weak.
> >
> > **b.**	Additionally, APO’s reliance on gradients and its iterative nature result in higher computational costs. Our experiments show that APO typically requires 2–3 hours for prompt optimization, with significantly higher API calls and token usage compared to PW, which achieves optimized prompts in just 20–30 minutes.
> >
> > 2. **PromptAgent vs. PW:**
> > PromptAgent introduces Monte Carlo Tree Search (MCTS) for exploring multiple prompt variations and uses an external LLM for feedback at every branch of the tree. While this enhances exploration, **feedback is not used to guide the exploration directly** but instead evaluates multiple independent variations. This results in inefficiency, as the system evaluates a large number of branches without iterative refinement driving convergence.
> >
> > **a.**	PW, on the other hand, uses feedback to **systematically guide exploration**, refining one best prompt at each iteration while integrating it with diverse examples. This approach ensures that feedback meaningfully shapes the optimization process, improving computational efficiency. PromptAgent often requires 3–4 hours for prompt generation, whereas PW completes the process in under 30 minutes, with superior performance demonstrated across datasets.
> >
> > 3. **PW’s Key Differentiators:**
> >
> > **a.	Dual Optimization:** PW optimizes both the prompt instruction and in-context examples in tandem, addressing a broader range of challenges than methods like APO and PromptAgent, which focus solely on instruction refinement.
> >
> > **b.	Feedback-Guided Exploration:** Unlike APO and PromptAgent, PW tightly integrates feedback into the exploration process, enabling more efficient and directed iterations.
> >
> > **c.	Efficiency and Versatility:** PW significantly reduces computational overhead while maintaining superior performance across a variety of complex datasets, as shown in our evaluations on GSM8k, AQUARAT, and others.
> >
> > We will incorporate this detailed discussion and corresponding quantitative comparisons into the revised manuscript to clarify PW’s novelty and advantages over existing feedback-based prompt optimization approaches. Thank you for highlighting this important aspect!
> >
> > >>Following above, the authors only compare with baselines like Evoprompt, PromptBreeder, and APE, which truly do not use error feedback. However, why not compare with works using error feedback such as APO? I am curious that these frameworks can achieve similar performance and low costs.
> >
> > Thank you for highlighting this concern. In response, we provide a detailed comparative analysis between PW and feedback-based approaches such as **APO** and **PromptAgent.**
> >
> > We evaluated **PW** against **APO** and **PromptAgent** across four datasets: **GSM8k, AQUARAT, SVAMP, and ETHOS** (see Table A below). The key findings are summarized below:
> >
> > 1.  **Accuracy**: PW consistently achieves the **highest accuracy** across all datasets, significantly outperforming APO and PromptAgent, especially on complex tasks such as **GSM8k**. For instance, PW demonstrates superior handling of reasoning-heavy datasets where feedback-guided iterative refinement is essential.
> >
> > 2.  **Efficiency**: PW demonstrates **significant efficiency** advantages over both APO and PromptAgent. By tightly integrating feedback into a **critique-and-synthesize loop**, PW minimizes redundant computations while systematically improving prompts.
> >
> > 3.  **Token Usage**: While APO produces simpler prompts with slightly lower token usage, this comes at the cost of significantly reduced accuracy. PW balances token efficiency with performance, ensuring minimal token consumption without sacrificing quality.
> >
> > We will incorporate this detailed analysis and corresponding results (e.g.,Table A below) in the revised manuscript.

---

> > > ### Author Response · Authors · 2024-11-22
> > > **Comparison with Feedback-based prompt optimization approaches: (2)**
> > >
> > > ### Table below shows the accuracy and cost analysis comparison between PW and other feedback-based methods.
> > >
> > > | Dataset | Accuracy(high) | Accuracy(high)|Accuracy(high) |Accuracy(high) | API Calls(low)| API Calls(low) |API Calls(low)  | API Calls(low)|  Tokens(low)| Tokens(low) | Tokens(low)|Tokens(low) |
> > > | ------- | --------------- | ----------- | ------------ | ------------ | ------------ | ------------ | ------------ |------------ |------------ |------------ |------------ |-----
> > > |  | DSPy    | PromptAgent     | APO         | PW           | DSPy | PromptAgent | APO | PW | DSPy | PromptAgent | APO | PW |
> > > | GSM8k   | 78.2            | 68.84       | 25.67        | **90** | 915 | 2115 | 8490 | **147** | 262 | 500 | **109** | 237 |
> > > | AQUARAT | 55.1            | 56.67       | 20.12        | **58.2** | 920 | 2200 | 8500 | **112** | 326 | 875 | **125** | 200|
> > > | SVAMP   | 77              | 78.67       | 75.25        | **82.3** | 2300 | 2111 | 8000 | **178** | 189 | 680 | **85** | 127 |
> > > | ETHOS   | 84.1            | 84.25       | 80.62        | **89.4** | 660 | 2217 | 8200 | **80** | 175 | 417 | **55** | 190 |
> > >
> > > >> The shared anonymized code link cannot be accessed.
> > >
> > > The shared anonymized code link has been tested and is accessible. Please try accessing it again. Additionally, the source code was included as a zip file in the supplementary materials. Kindly check the supplementary file and let us know if you continue to face issues.
> > >
> > > >> The specific setting of baseline methods is not illustrated. How to make sure a fair comparison like the same number of prompt sampling?
> > >
> > > Thank you for raising this concern. For all baseline methods, we used the best hyper-parameter settings provided in their respective codebases or reported in their papers, ensuring optimal performance for each. This approach guarantees a fair comparison by evaluating every baseline under its recommended configuration, including prompt sampling numbers and other key parameters. Altering these parameters could lead to suboptimal results for baselines, so we adhered strictly to their default best settings to ensure a robust and equitable evaluation against PW. We hope this clarifies our methodology.
> > >
> > > >>In the optimization of few-shot examples, how to make sure the examples do not contain errors if relying on LLMs to synthesize?
> > >
> > > Thank you for the insightful question. As described in Section 3.4, the synthesized examples undergo a self-evaluation process using the LLM as part of the validation component. Specifically, the validation component leverages the LLM to assess the coherence, correctness, and relevance of the examples with respect to the task. This ensures that any incorrect or irrelevant examples are filtered out before being included in the final prompt. This mechanism helps maintain the integrity and quality of the optimized few-shot examples.
> > >
> > > >>The framework relies heavily on LLM capability to refine and reflect. Is there any components that smaller LLMs cannot work well so that the framework fails?
> > >
> > > Thank you for the insightful question. As discussed in Section 6.2, we evaluate PW's performance with smaller LLMs and present findings in Table 6. For instance, prompts generated by PW using LLaMA-70B show a negligible accuracy drop (<1%) compared to those generated by GPT-4, highlighting the framework's robustness with moderately sized models. However, experiments with smaller models like LLaMA-3-8B reveal significant performance degradation.
> > >
> > > The primary reasons for this decline are:
> > >
> > > 1.  **Ineffective Feedback Generation**: Smaller models often struggle to provide precise and actionable feedback, which is crucial for refining prompts.
> > > 2.  **Inadequate Synthesis Capability**: These models lack the ability to synthesize coherent and high-quality prompts or examples effectively, impacting the optimization process.

---

> > > > ### Author Response · Authors · 2024-11-22
> > > > **Robust Ablation studies**
> > > >
> > > > >>Many components lack analysis of why it works and lack the study over the robustness of this framework. For example, whether the wording of component prompt will influence the performance.
> > > >
> > > > We included an ablation study in the original paper but recognize that more detailed analyses strengthen the framework's robustness. To this end, we conducted extensive ablation studies on GSM8k (2-shot setting), demonstrating the complementary roles of stage-wise optimization, synthesized examples, and reasoning steps. Results show the lowest accuracy with Stage 1 alone, gradual improvement with Stage 2, and the best performance when combining all components, including synthesis and reasoning.
> > > >
> > > > | **Experiment**                         | **Description**                                              | **Accuracy** |
> > > > |-----------------------------------------|--------------------------------------------------------------|--------------|
> > > > | **Expt 1**                              | Stage 1: Only prompt optimization                            | 0.52         |
> > > > | **Expt 2**                              | Stage 2: Joint optimization without examples (zero-shot)     | 0.65         |
> > > > | **Expt 3**                              | Stage 1 + Stage 2 without examples (zero-shot)               | 0.73         |
> > > > | **Expt 4**                              | Stage 2 with examples (no synthesis)                         | 0.70         |
> > > > | **Expt 5**                              | Stage 1 + Stage 2 with examples (no synthesis)               | 0.77         |
> > > > | **Expt 6**                              | Stage 2 with synthesized examples                            | 0.71         |
> > > > | **Expt 7**                              | Stage 1 + Stage 2 with synthesized examples                  | 0.79         |
> > > > | **Expt 8**                              | Stage 1 + Stage 2 + Examples (no synthesis) + CoT reasoning  | 0.80         |
> > > > | **Expt 9**                              | Stage 1 + Stage 2 + Examples (with synthesis) + CoT reasoning| 0.83         |

---

> > > > > ### Author Response · Authors · 2024-11-25
> > > > >
> > > > > Thank you again for your feedback and thoughtful questions! As we are now approaching the end of the rebuttal period, I’m following up again to check if you have any further questions regarding our responses. We hope we have addressed your concerns and clarifications thoroughly and looking forward to a positive response and score.

---

> > > > > > ### Author Response · Authors · 2024-11-25
> > > > > >
> > > > > > Thank you again for your feedback and thoughtful questions! As we are now approaching the end of the rebuttal period, I’m following up again to check if you have any further questions regarding our responses. We hope we have addressed your concerns and clarifications thoroughly and looking forward to a positive response and score.

---

> > > > > > > ### Comment · Reviewer_hKKu · 2024-11-27
> > > > > > > **Thanks for the detailed responses**
> > > > > > >
> > > > > > > Hello authors,
> > > > > > >
> > > > > > > Thank you for your detailed responses. Some of my questions have been addressed. I appreciate the detailed comparison with other methods and added ablation studies. However, based on my questions and questions from other reviewers, I think this paper has too many parts needed to be enhanced. Meanwhile, why combining these multiple parts together can work remains a question. Is it purely empirical without any theoretical analysis and insights? In this regard, I will keep my overall evaluation.
> > > > > > >
> > > > > > > Best,
> > > > > > > Reviewer

---

### Official Review · Reviewer_SXFP · 2024-11-03

**Soundness:** 4
**Presentation:** 3
**Contribution:** 3
**Rating:** 5
**Confidence:** 4

**Summary:**

This paper presents a new discrete prompt optimization method that automatically refines both prompt instructions and in-context examples by using LLM.

**Strengths:**

1. The method PW proposed in this paper optimizes both the prompt instructions and in-context examples contained in the discrete prompt, which achieves a comprehensive approach and considerable performance gains in experiments.
2. Compared to recent LLM-based prompt optimization methods, PW achieves satisfying results with much less cost of API calls.

**Weaknesses:**

1. Although the authors have written about the difference between PW and existing prompt optimization methods. The technical novelty of the proposed method is still somewhat weak. The approach of refining prompt instructions and selecting the most effective in-context examples are widely recognized concepts.
2. The presentation of the overall pipeline of the approach could be improved. For example, moving the main algorithm in the appendix into the main text will make the presentation clearer and avoid making readers lose among the proposed components.

**Questions:**

Please respond to the concerns in the "Weaknesses" part.

Q1. Regarding the calculation of the API cost of PW, how do you evaluate the performance of the current prompt instructions and selected in-context examples with such a small quantity of API calls? Because stable evaluation results of the prompts are a prerequisite for stable optimization.

Q2. What are the ablation results of solely optimizing prompt instructions and in-context examples?

Q3. Line 252: What "scoring mechanism"?

Q4. Line 256: In what case will "this process does not yield the desired count"?

---

> ### Author Response · Authors · 2024-11-22
>
> Thank you for your insightful review and positive feedback. Below are our responses to address your concerns:
>
> >>Although the authors have written about the difference between PW and existing prompt optimization methods. The technical novelty of the proposed method is still somewhat weak. The approach of refining prompt instructions and selecting the most effective in-context examples are widely recognized concepts.
>
> PromptWizard (PW) achieves its superior performance through three core innovations that collectively differentiate it from existing approaches, ensuring both high accuracy and cost-efficiency. These insights address the limitations of prior work and offer a transferable foundation for future advancements:
>
> 1. **Feedback-Driven Critique-and-Synthesis Mechanism:**
> PW introduces a novel feedback loop where an LLM generates, critiques, and refines prompts and examples iteratively. This mechanism balances **exploration** (through diverse mutations) and **exploitation** (guided feedback to converge on optimal prompts). Unlike evolutionary approaches such as PromptBreeder and EvoPrompt, which rely on mutation without structured feedback—leading to uncontrolled exploration and higher costs—PW integrates LLM-driven feedback at every stage to guide prompt refinement. This feedback loop ensures edits align with task requirements, reducing unnecessary evaluations and delivering better results with significantly lower computational costs. Compared to feedback-based systems like APO and PromptAgent, which lack diverse initial mutations, PW expands the search space by generating diverse prompts before applying feedback, preventing premature convergence to suboptimal solutions. This balanced and iterative process ensures task-relevant optimization while reducing computational overhead.
>
> 2.  **Sequential Optimization of Instructions and Examples:**
> PW uniquely optimizes instructions and examples in tandem through iterative refinement:
>
> a.  **Prompt Optimization with Examples:** The refined prompt undergoes evaluation with selected examples, and feedback informs further improvements.
>
> b.	**Example Synthesis:** Using feedback, PW synthesizes diverse, task-relevant examples that enhance performance.
> By jointly optimizing prompts and examples, PW surpasses methods that treat these elements independently, creating cohesive, task-aware prompts.
>
> 3. **Incorporation of CoT Reasoning:**
> PW integrates Chain-of-Thought (CoT) reasoning during optimization. This step-by-step reasoning framework leverages the refined prompt to guide the LLM in solving tasks effectively, further improving task performance and interpretability.
>
> ### Takeaway
> The key insight is that PW combines feedback-driven critique and synthesis, tandem optimization of prompts and examples, and self-generated reasoning steps into an automated, efficient framework. These elements are fundamental to PW's ability to outperform state-of-the-art methods while being significantly more cost-effective.
> While certain components may seem familiar, PW innovates by integrating these techniques into a cohesive system that automates and optimizes prompt engineering effectively. This work lays the foundation for future developments in prompt optimization by demonstrating that these techniques can be adapted and extended to address evolving challenges in this field.
>
> >>The presentation of the overall pipeline of the approach could be improved. For example, moving the main algorithm in the appendix into the main text will make the presentation clearer and avoid making readers lose among the proposed components.
>
> Thank you for this suggestion. In the revised manuscript, we will move the main algorithm from the appendix to the main text. This adjustment will ensure that readers can follow the proposed components more easily and see how they integrate into the overall pipeline.

---

> > ### Author Response · Authors · 2024-11-22
> >
> > >> Q1. Regarding the calculation of the API cost of PW, how do you evaluate the performance of the current prompt instructions and selected in-context examples with such a small quantity of API calls? Because stable evaluation results of the prompts are a prerequisite for stable optimization.
> >
> > Thank you for your question. We believe there might be some confusion regarding the API cost analysis. Let us clarify the distinction between prompt generation and evaluation, as detailed in the paper:
> >
> > 1. **Prompt Generation Costs (Phase 1):**
> > As outlined in Section 5.2 (Table 4), the cost analysis focuses on the prompt generation phase. This phase involves using API calls to iteratively refine the prompt and in-context examples. PromptWizard demonstrates significant cost efficiency in this phase, requiring far fewer API calls and tokens compared to other evolutionary and feedback-based approaches. This efficiency is primarily achieved through PromptWizard's three key building blocks: (i) feedback-driven critique-and-synthesis, (ii) sequential optimization of instructions and examples, and (iii) effective use of CoT reasoning.
> >
> > 2. **Evaluation Costs (Phase 2):**
> > Once the optimized prompt is generated during Phase 1, it is then evaluated across the test set. For this phase, **all prompt optimization techniques**, including PromptWizard, require just one API call per test sample. For example, on test sets with 1,000+ questions, each prompt is used as-is, without further modification, to evaluate its effectiveness.
> >
> > 3. **Clarifying the Results in Section 5.2:**
> > The results presented in Table 4 pertain exclusively to the prompt generation phase and not the evaluation phase. PromptWizard's low API costs reflect its efficiency in refining the prompt during Phase 1. The evaluation phase does not involve iterative optimization; instead, the final optimized prompt is directly applied to all test samples.
> >
> > We hope this distinction between prompt generation and evaluation phases clarifies your concerns.
> >
> > >> Q2. What are the ablation results of solely optimizing prompt instructions and in-context examples?
> >
> > We conducted detailed experiments isolating key building blocks of PromptWizard (PW) to evaluate their individual contributions. These ablations reinforce the significance of PW’s modular components—stage-wise optimization, synthesis of examples, and reasoning steps—highlighting their complementary roles in achieving superior task performance.
> >
> > The experiments, performed on the GSM8k dataset with a 2-shot setting, illustrate the impact of stage-wise optimization, synthesized examples, and reasoning steps. Results show that **Stage 1 (only prompt optimization)** achieves the lowest accuracy. Adding **Stage 2 (joint optimization of prompts and examples)** slightly improves results. The best performance is achieved by combining all components, including synthesized examples and reasoning steps, which highlight their complementary roles.
> >
> > We will include these results in the revised manuscript for further clarity. The detailed ablation results are provided below:
> > | **Experiment**                         | **Description**                                              | **Accuracy** |
> > |-----------------------------------------|--------------------------------------------------------------|--------------|
> > | **Expt 1**                              | Stage 1: Only prompt optimization                            | 0.52         |
> > | **Expt 2**                              | Stage 2: Joint optimization without examples (zero-shot)     | 0.65         |
> > | **Expt 3**                              | Stage 1 + Stage 2 without examples (zero-shot)               | 0.73         |
> > | **Expt 4**                              | Stage 2 with examples (no synthesis)                         | 0.70         |
> > | **Expt 5**                              | Stage 1 + Stage 2 with examples (no synthesis)               | 0.77         |
> > | **Expt 6**                              | Stage 2 with synthesized examples                            | 0.71         |
> > | **Expt 7**                              | Stage 1 + Stage 2 with synthesized examples                  | 0.79         |
> > | **Expt 8**                              | Stage 1 + Stage 2 + Examples (no synthesis) + CoT reasoning  | 0.80         |
> > | **Expt 9**                              | Stage 1 + Stage 2 + Examples (with synthesis) + CoT reasoning| 0.83         |
> >
> >
> >
> > >> Q4 Line 256: In what case will "this process does not yield the desired count"?
> >
> > This situation can occur when the prompt, either the initial or the refined version after feedback-driven improvement, performs well from the start, leading to all 25 selected examples being classified as positive. In this case, there are no negative samples, meaning the prompt already produces optimal or near-optimal results for all selected examples.

---

> > > ### Author Response · Authors · 2024-11-22
> > >
> > > >> Q3 Line 252: What "scoring mechanism"?
> > >
> > > PromptWizard uses a scoring mechanism to evaluate the performance of the current prompt against selected examples. The mechanism assesses how effectively the prompt generates correct responses relative to the ground truth. The scoring mechanism can be either using traditional metrics like accuracy, F1 score or an LLM as an evaluator that evaluates the correctness and relevance of generated responses against the ground truth. The scores are employed to classify the selected examples into **positive** (well-performing examples) and **negative** (poorly performing examples) categories. This classification informs the critique-and-synthesis mechanism, allowing for the systematic refinement of both the prompts and examples, ensuring better task performance.
> > >
> > > **We hope our responses have thoroughly addressed all concerns, providing clarity and eliminating any ambiguities. **

---

> > > > ### Author Response · Authors · 2024-11-25
> > > >
> > > > Thank you again for your feedback and thoughtful questions! As we are now approaching the end of the rebuttal period, I’m following up again to check if you have any further questions regarding our responses. We hope we have addressed your concerns and clarifications thoroughly and looking forward to a positive response and score.

---

> > > > > ### Author Response · Authors · 2024-11-25
> > > > >
> > > > > Thank you again for your feedback and thoughtful questions! As we are now approaching the end of the rebuttal period, I’m following up again to check if you have any further questions regarding our responses. We hope we have addressed your concerns and clarifications thoroughly and looking forward to a positive response and score.

---

> > > > > > ### Author Response · Authors · 2024-11-29
> > > > > >
> > > > > > Thank you again for your feedback and thoughtful questions! As we are now approaching the end of the rebuttal period, I’m following up again to check if you have any further questions regarding our responses. We hope we have addressed your concerns and clarifications thoroughly and looking forward to a positive response and score.

---

> > > > > > > ### Author Response · Authors · 2024-12-02
> > > > > > >
> > > > > > > Thank you again for your feedback and thoughtful questions! As we are now approaching the end of the rebuttal period, I’m following up again to check if you have any further questions regarding our responses. We hope we have addressed your concerns and clarifications thoroughly and looking forward to a positive response and score.

---

### Official Review · Reviewer_mroo · 2024-11-03

**Soundness:** 3
**Presentation:** 3
**Contribution:** 2
**Rating:** 6
**Confidence:** 3

**Summary:**

PromptWizard explores automated discrete prompt engineering by combining a genetic search process with a critique and refinement stage.  In addition, PromptWizard also adds a few-shot reasoning chain for the selected prompts.  They implemented what seems like a fairly large system and evaluated it on the BigBench Instruction Induction Benchmark, GSM8K, AQUARAT, and SVAMP.  They show significant wins in accuracy, cost, and efficiency other published systems including Instinct, InstructZero, PromptBreeder, EvoPrompt and AVE.

**Strengths:**

1. The proposed system appears pretty extensive and combines several techniques for discrete prompt engineering.
1. The experimental evaluation is well done and the results show significant wins in accuracy and cost over existing systems.
1. The ablation points to potentially significant wins from their method to construct few-shot chain-of-thought reasoning examples.

**Weaknesses:**

1. My primary concern with this paper is that while the results are impressive, I am struggling to identify the key-insight or transferable idea.  What makes PromptWizard better than all the systems it beats? There is an ablation but it is one paragraph and seems to suggest that reasoning and few shot examples is primary source of improvement.
2. While the writing was reasonable, the fairly complex pipeline of evolutionary optimizers and sequential optimization made it difficult to understand how the two approaches work together.  Are they really just two sequential stages? Shouldn't the critical feedback be used in the iterative optimization.
3. Given the incorporation of few-shot examples, I was also expecting see comparisons with DSPy but this didn't appear anywhere in the paper.

**Questions:**

1. What is the big idea I should take away from this paper?
1. How does the sequential optimization interact with the iterative optimization? Is it really just a two stage pipeline?
1. How does this work compare with DSPy?

---

> ### Author Response · Authors · 2024-11-22
>
> Thank you for your insightful review and positive feedback. We appreciate your suggestions, which will enhance the manuscript. Below are our responses to address your concerns:
>
> >> My primary concern with this paper is that while the results are impressive, I am struggling to identify the key-insight or transferable idea. What makes PromptWizard better than all the systems it beats?
> What is the big idea I should take away from this paper?
>
> PromptWizard (PW) achieves its superior performance through three core innovations that collectively differentiate it from existing approaches, ensuring both high accuracy and cost-efficiency. These insights address the limitations of prior work and offer a transferable foundation for future advancements:
>
> 1. **Feedback-Driven Critique-and-Synthesis Mechanism:**
> PW introduces a novel feedback loop where an LLM generates, critiques, and refines prompts and examples iteratively. This mechanism balances **exploration** (through diverse mutations) and **exploitation** (guided feedback to converge on optimal prompts). Unlike evolutionary approaches such as PromptBreeder and EvoPrompt, which rely on mutation without structured feedback—leading to uncontrolled exploration and higher costs—PW integrates LLM-driven feedback at every stage to guide prompt refinement. This feedback loop ensures edits align with task requirements, reducing unnecessary evaluations and delivering better results with significantly lower computational costs. Compared to feedback-based systems like APO and PromptAgent, which lack diverse initial mutations, PW expands the search space by generating diverse prompts before applying feedback, preventing premature convergence to suboptimal solutions. This balanced and iterative process ensures task-relevant optimization while reducing computational overhead.
>
> 2.  **Sequential Optimization of Instructions and Examples:**
> PW uniquely optimizes instructions and examples in tandem through iterative refinement:
>
> a.  **Prompt Optimization with Examples:** The refined prompt undergoes evaluation with selected examples, and feedback informs further improvements.
>
> b.	**Example Synthesis:** Using feedback, PW synthesizes diverse, task-relevant examples that enhance performance.
> By jointly optimizing prompts and examples, PW surpasses methods that treat these elements independently, creating cohesive, task-aware prompts.
>
> 3. **Incorporation of CoT Reasoning:**
> PW integrates Chain-of-Thought (CoT) reasoning during optimization. This step-by-step reasoning framework leverages the refined prompt to guide the LLM in solving tasks effectively, further improving task performance and interpretability.
>
> ### Takeaway
> The key insight is that PW combines feedback-driven critique and synthesis, tandem optimization of prompts and examples, and self-generated reasoning steps into an automated, efficient framework. These elements are fundamental to PW's ability to outperform state-of-the-art methods while being significantly more cost-effective.
> While certain components may seem familiar, PW innovates by integrating these techniques into a cohesive system that automates and optimizes prompt engineering effectively. This work lays the foundation for future developments in prompt optimization by demonstrating that these techniques can be adapted and extended to address evolving challenges in this field.
>
> >>While the writing was reasonable, the fairly complex pipeline of evolutionary optimizers and sequential optimization made it difficult to understand how the two approaches work together..
> >> How does the sequential optimization interact with the iterative optimization? Is it really just a two stage pipeline?
>
> Thank you for your insightful comment. We acknowledge that the complexity of the pipeline might have caused some confusion, and we will revise our writing to make it clearer. PromptWizard (PW) is indeed a two-stage sequential pipeline where feedback-driven critique-and-synthesis plays a central role in both stages:
>
> 1.  **Stage 1 (Iterative Prompt Optimization):** This stage focuses solely on refining the prompt instruction. Using an evolutionary approach, diverse initial prompts are generated and iteratively refined through feedback-driven critique-and-synthesis to converge on an optimized prompt.
> 2.  **Stage 2 (Sequential Optimization of Prompt and Examples):** In this stage, the refined prompt from Stage 1 is paired with selected examples, and both are optimized together. The critique-and-synthesis mechanism ensures alignment between the examples and the prompt while synthesizing new examples for improved task performance.
>
> The critical feedback loop ensures that both stages iteratively refine their respective components, yet the overall process follows a sequential structure: prompt optimization first (Stage 1), followed by joint optimization of the prompt and examples (Stage 2). We will clarify this interaction further in the revision to ensure accessibility and coherence.

---

> > ### Author Response · Authors · 2024-11-22
> >
> > **Additional Ablation Studies:**
> >
> > We have conducted detailed experiments isolating key building blocks of PromptWizard (PW) to evaluate their individual contributions. These ablations reinforce the significance of PW’s modular components—stage-wise optimization, synthesis of examples, and reasoning steps—highlighting their complementary roles in achieving superior task performance.
> >
> > The experiments, performed on the GSM8k dataset with a 2-shot setting, illustrate the impact of stage-wise optimization, synthesized examples, and reasoning steps. Results show that **Stage 1 (only prompt optimization)** achieves the lowest accuracy. Adding **Stage 2 (joint optimization of prompts and examples)** slightly improves results. The best performance is achieved by combining all components, including synthesized examples and reasoning steps, which highlight their complementary roles.
> >
> > We will include these results in the revised manuscript for further clarity. The detailed ablation results are provided below:
> >
> > | **Experiment**                         | **Description**                                              | **Accuracy** |
> > |-----------------------------------------|--------------------------------------------------------------|--------------|
> > | **Expt 1**                              | Stage 1: Only prompt optimization                            | 0.52         |
> > | **Expt 2**                              | Stage 2: Joint optimization without examples (zero-shot)     | 0.65         |
> > | **Expt 3**                              | Stage 1 + Stage 2 without examples (zero-shot)               | 0.73         |
> > | **Expt 4**                              | Stage 2 with examples (no synthesis)                         | 0.70         |
> > | **Expt 5**                              | Stage 1 + Stage 2 with examples (no synthesis)               | 0.77         |
> > | **Expt 6**                              | Stage 2 with synthesized examples                            | 0.71         |
> > | **Expt 7**                              | Stage 1 + Stage 2 with synthesized examples                  | 0.79         |
> > | **Expt 8**                              | Stage 1 + Stage 2 + Examples (no synthesis) + CoT reasoning  | 0.80         |
> > | **Expt 9**                              | Stage 1 + Stage 2 + Examples (with synthesis) + CoT reasoning| 0.83         |
> >
> >
> > **Comparison with DSPy:**
> > >> Given the incorporation of few-shot examples, I was also expecting see comparisons with DSPy but this didn't appear anywhere in the paper.
> > >> How does this work compare with DSPy?
> >
> > Thank you for raising this point. We conducted direct comparisons between PromptWizard (PW) and DSPy on GSM8k, AQUA-RAT, SVAMP, and Ethos datasets, observing that PW consistently outperforms DSPy in accuracy while being significantly more computationally efficient in terms of LLM calls and average token usage for both input and output. These results, which we will include in the main paper, highlight PW's efficiency and effectiveness.
> >
> > While both approaches optimize prompt instructions and examples, PW introduces several distinctions. Unlike DSPy, which uses CoT, ReAct, and other strategies as foundational blocks for prompt generation—often leading to higher computational costs—PW leverages a feedback-driven critique-and-synthesis process to iteratively and synergistically optimize instructions and examples in tandem. This dynamic refinement, guided by task-specific feedback, enables PW to achieve superior performance without incurring the high costs associated with DSPy’s modules.
> >
> > We appreciate the opportunity to clarify this and will ensure these comparisons are included in the revised manuscript.
> > | Dataset | Accuracy(high) |Accuracy(high) |API Calls(low) |API Calls(low) |Tokens(low)  | Tokens(low)  |
> > | ------- | --------------- | ----------- | ------------ | ------------ | ------------ | ----------------
> > |  | DSPy    |  PW           | DSPy | PW | DSPy |PW |
> > | GSM8K     | 78    | **90**      | 915   | **147**     | 262   | **237**   |
> > | AQUARAT   | 55    | **58**      | 920   | **112**     | 326   | **200**   |
> > | SVAMP     | 77    | **82**      | 2300  | **178**     | 189   | **127**   |
> > | ETHOS     | 84    | **89**      | 660   | **80**      | **176**   | 190   |
> >
> > **We hope our responses have thoroughly addressed all concerns, providing clarity and eliminating any ambiguities. We are happy to provide additional clarifications or experimental results as required.**

---

> > > ### Author Response · Authors · 2024-11-25
> > >
> > > Thank you again for your feedback and thoughtful questions! As we are now approaching the end of the rebuttal period, I’m following up again to check if you have any further questions regarding our responses. We hope we have addressed your concerns and clarifications thoroughly and looking forward to a positive response and score.

---

> > > > ### Author Response · Authors · 2024-11-25
> > > >
> > > > Thank you again for your feedback and thoughtful questions! As we are now approaching the end of the rebuttal period, I’m following up again to check if you have any further questions regarding our responses. We hope we have addressed your concerns and clarifications thoroughly and looking forward to a positive response and score.

---

> > > > > ### Author Response · Authors · 2024-11-29
> > > > >
> > > > > Thank you again for your feedback and thoughtful questions! As we are now approaching the end of the rebuttal period, I’m following up again to check if you have any further questions regarding our responses. We hope we have addressed your concerns and clarifications thoroughly and looking forward to a positive response and score.

---

> > > > > > ### Author Response · Authors · 2024-12-02
> > > > > >
> > > > > > Thank you again for your feedback and thoughtful questions! As we are now approaching the end of the rebuttal period, I’m following up again to check if you have any further questions regarding our responses. We hope we have addressed your concerns and clarifications thoroughly and looking forward to a positive response and score.

---

### Author Response · Authors · 2024-11-23

We thank all the reviewers for their service in reviewing our work, PromptWizard, and for their detailed and insightful comments. We have clarified all the concerns raised by the reviewer and providing a brief summary below:

## PromptWizard Contributions:

PromptWizard achieves superior performance and cost-efficiency through three key innovations:

1. **Feedback-Driven Critique-and-Synthesis**

*   Iterative feedback loop for generating, critiquing, and refining prompts.
*   Balances diverse exploration with guided optimization, reducing costs.
*   Outperforms evolutionary (e.g., PromptBreeder) and feedback-based (e.g., APO) methods by integrating structured feedback at every step.

2.  **Joint Optimization of Prompts and Examples**

*   Simultaneously refines instructions and task-relevant examples.
*   Creates cohesive, context-aware prompts, outperforming methods that optimize these elements separately.

3.   **Incorporation of Chain-of-Thought (CoT) Reasoning**

*   Enhances interpretability and accuracy through structured, step-by-step reasoning integrated into the framework.

**Key Differentiators**

*   **Over Evolutionary Algorithms (e.g., PromptBreeder):** Combines feedback-driven optimization with diverse exploration, avoiding uncontrolled or random prompt generation.
*   **Over Feedback-Based Systems (e.g., APO, PromptAgent):** Tightly integrates feedback with exploration, optimizing both prompts and examples for higher efficiency and accuracy.

**Takeaway**

PromptWizard delivers a scalable, automated framework combining feedback-driven synthesis, tandem optimization, and CoT reasoning. It sets a new benchmark for accuracy and cost-effectiveness, addressing limitations of existing methods while paving the way for future advancements in prompt optimization.

## Comparison with feedback-based approaches:

Below is a detailed articulation of how PromptWizard (PW) stands apart from these methods, highlighting its unique contributions and superior performance:

**1\. APO vs. PW:**

*   **APO’s Limitations:** Relies on gradient-based feedback and bandit algorithms, lacks diverse exploration, and incurs high computational costs (2–3 hours per optimization).
*   **PW’s Advantages:** Introduces a feedback-driven critique-and-synthesis mechanism with diverse reasoning styles, ensuring robustness and efficiency (20–30 minutes for optimization).

**2\. PromptAgent vs. PW:**

*   **PromptAgent’s Limitations:** Uses Monte Carlo Tree Search (MCTS) for prompt exploration but treats feedback as an evaluator, leading to inefficiency (3–4 hours per optimization).
*   **PW’s Advantages:** Leverages feedback to guide iterative refinement, combining it with diverse examples for faster optimization (<30 minutes) and better performance.

| Dataset | Accuracy(high) | Accuracy(high)|Accuracy(high) |Accuracy(high) | API Calls(low)| API Calls(low) |API Calls(low)  | API Calls(low)|  Tokens(low)| Tokens(low) | Tokens(low)|Tokens(low) |
| ------- | --------------- | ----------- | ------------ | ------------ | ------------ | ------------ | ------------ |------------ |------------ |------------ |------------ |-----
|  | DSPy    | PromptAgent     | APO         | PW           | DSPy | PromptAgent | APO | PW | DSPy | PromptAgent | APO | PW |
| GSM8k   | 78.2            | 68.84       | 25.67        | **90** | 915 | 2115 | 8490 | **147** | 262 | 500 | **109** | 237 |
| AQUARAT | 55.1            | 56.67       | 20.12        | **58.2** | 920 | 2200 | 8500 | **112** | 326 | 875 | **125** | 200|
| SVAMP   | 77              | 78.67       | 75.25        | **82.3** | 2300 | 2111 | 8000 | **178** | 189 | 680 | **85** | 127 |
| ETHOS   | 84.1            | 84.25       | 80.62        | **89.4** | 660 | 2217 | 8200 | **80** | 175 | 417 | **55** | 190 |
## Additional ablation results

We conducted ablations on PromptWizard (PW) to assess the contributions of its key components: stage-wise optimization, example synthesis, and reasoning steps. Experiments on GSM8k (2-shot setting) reveal that Stage 1 (prompt optimization alone) yields the lowest accuracy. Adding Stage 2 (joint optimization of prompts and examples) improves performance, while combining all components—including synthesized examples and reasoning steps—achieves the best results, underscoring their complementary roles.

**Our clarifications, along with additional results, demonstrate the effectiveness and generalization of PromptWizard across many tasks and domains.**

---

### Meta-Review · Area_Chair_b2Ln · 2024-12-25

**Metareview:**

This paper proposed PromptWizard, a new prompt optimization algorithm which considers feedbacks and joint optimization over instructions and few-shots examples to iteratively generate and refine the prompts. PromptWizard demonstrates superior performance and cost-efficiency in many LLM tasks. However, reviewers have raised concerns regarding the novelty and clarity of the paper. To address this, I recommend enhancing the presentation and providing further clarification to better highlight the unique contributions of PromptWizard.

**Additional Comments On Reviewer Discussion:**

The main concerns raised by the reviewers are 1) novelty; 2) ablations; 3) comparison with other feedback based prompt optimization algorithms. In the rebuttal, new ablations and comparison with other feedback based prompt optimization algorithms are provided. However it is still a question whether the work is novel enough.

---

### Decision · Program_Chairs · 2025-01-22

Reject